# Brief Electrical Stimulation Promotes Recovery after Surgical Repair of Injured Peripheral Nerves

**DOI:** 10.3390/ijms25010665

**Published:** 2024-01-04

**Authors:** Tessa Gordon

**Affiliations:** Division of Reconstructive Surgery, Department of Surgery, University of Toronto, Toronto, ON M4G 1X8, Canada; tessa.gordon@sickkids.ca

**Keywords:** peripheral nerve regeneration, delayed surgical nerve repair, regeneration-associated genes, electrical stimulation

## Abstract

Injured peripheral nerves regenerate their axons in contrast to those in the central nervous system. Yet, functional recovery after surgical repair is often disappointing. The basis for poor recovery is progressive deterioration with time and distance of the growth capacity of the neurons that lose their contact with targets (chronic axotomy) and the growth support of the chronically denervated Schwann cells (SC) in the distal nerve stumps. Nonetheless, chronically denervated atrophic muscle retains the capacity for reinnervation. Declining electrical activity of motoneurons accompanies the progressive fall in axotomized neuronal and denervated SC expression of regeneration-associated-genes and declining regenerative success. Reduced motoneuronal activity is due to the withdrawal of synaptic contacts from the soma. Exogenous neurotrophic factors that promote nerve regeneration can replace the endogenous factors whose expression declines with time. But the profuse axonal outgrowth they provoke and the difficulties in their delivery hinder their efficacy. Brief (1 h) low-frequency (20 Hz) electrical stimulation (ES) proximal to the injury site promotes the expression of endogenous growth factors and, in turn, dramatically accelerates axon outgrowth and target reinnervation. The latter ES effect has been demonstrated in both rats and humans. A conditioning ES of intact nerve days prior to nerve injury increases axonal outgrowth and regeneration rate. Thereby, this form of ES is amenable for nerve transfer surgeries and end-to-side neurorrhaphies. However, additional surgery for applying the required electrodes may be a hurdle. ES is applicable in all surgeries with excellent outcomes.

## 1. Introduction

Romon Y Cajal [1] recognized the contrasting capacity of injured nerves to regenerate in the peripheral nervous system (PNS) and not in the central nervous system (CNS). Yet, recovery of function is generally disappointing despite the contrasting support of the regrowth (regeneration) of the nerve fibres by the SCs in the PNS and the lack of support by the oligodendrocytes in the CNS [2,3]. Fewer than 50% of patients regain adequate motor or sensory function after surgical repair of injured median or ulnar nerves [4]. Indeed, only ~10% of the two million Americans suffering some form of peripheral nerve injury recover function; many having impaired motor and sensory function and frequently suffering pain [5]. Functional recovery varies with location and severity. The most severe are the more proximal nerve injuries [6] with brachial plexus injury being the most disabling [7]. Hand function is restored in only 1.2% of patients with multiple-traumatic injuries [7]. Typically healthy, young, economically productive adult patients need long periods of rehabilitation, and many must make career changes [8].

The relatively neglected issue of poor functional recovery has been addressed by the administration of drugs such as FK506 [9]. The efficacy of local FK506 delivery that has been demonstrated in *in vivo* and *in vitro* animal studies [9,10,11,12,13,14,15] and its potential clinical use has been reviewed recently [10] and hence, will not be considered further here. Electrical stimulation (ES) of muscles is a standard manipulation by physiotherapists aiming to prevent denervation atrophy and joint fixation [16,17]. In contrast, ES for injured nerves has been a more recent approach to promote muscle and sensory reinnervation [18,19,20,21,22,23,24,25], and these findings of Brushart and Gordon are largely confirmed by many investigators [26,27,28,29,30,31,32,33,34,35,36,37,38,39,40,41,42,43,44,45,46,47,48,49,50]. This review considers (1) how and why functional recovery is poor after peripheral nerve injury, and (2) the efficacy of neurotrophic factors and/or brief low-frequency ES to counteract the negative effects of delayed surgical repair as a prelude to advocate ES to promote nerve regeneration and functional recovery after human nerve injuries. Previous reviews have concerned one or more of these issues [9,51,52,53,54,55,56,57,58,59,60,61].

## 2. Peripheral Nerve Injury

After crush (axonotmesis) or transection (neurotmesis) nerve injuries, the neurons and their nerve fibres proximal to the injury are separated from their peripheral targets, the state of axotomy [51,62]. The growth-associated changes in the axotomized neurons and denervated SCs in the distal nerve stumps that are precursors to nerve regeneration and target reinnervation are presented prior to considering their decline with time after and distance from nerve injury.

### 2.1. Chromatolysis and Gene Expression in Axotomized Neurons

Chromatolysis refers to the classical morphological changes in axotomized neurons of the movement of the nucleus to an eccentric position and Nissl body dispersion in the somal cytoplasm [63]. The changes reflect increased neuronal metabolism and protein synthesis [64] as the neurons transition from normal transmitting to a growth/regenerative state [65]. This transition is usually driven by the transcriptome in which transcription factors coordinate the expression of multiple regeneration-associated genes (RAGs; [66,67]; see Section 2.2). Genes that translate proteins for neurotransmitter synthesis are downregulated whilst RAGs are upregulated (Figure 1A; [51,65]). Upregulated cytoskeletal proteins, tubulin and actin, are essential for transporting materials from the soma to the growth cones for elongation of the regenerating axons [68,69,70]. The corresponding downregulation of neurofilament protein that controls the axon calibre in peripheral nerves [51,68,71] accounts for the reduced size of nerves proximal to the injury site, as measured electrophysiologically [72] and morphologically [71,73]. The expression of neurotrophic factors, including *BDNF* and its receptors *trkB* and *p75NTR*, is upregulated in motoneurons (Figure 1A; [51,52,74,75]). Several phenotypes of the heterogeneous DRG sensory neurons and their Ia and Ib muscle and tendon afferent nerve fibres respectively, and their cutaneous afferent nerve fibres supplying the skin, transition to a more homogenous phenotype after axotomy [76].

### 2.2. Neuronal Molecular Signaling of Nerve Injury

Neuronal signaling of peripheral nerve injury occurs in a rapid phase dictated by a retrograde calcium wave to the neuronal soma and a later slow signaling phase, characterized by the retrograde transport of signaling molecules by motor proteins [77,78].

In the first rapid phase, disruption of the axonal membrane at the site of the nerve injury exposes the cytoplasm to external ionic concentrations. Within seconds, calcium ions enter the proximal nerve stump from the external fluid and the intracellular calcium concentration rises as membrane depolarization activates voltage-gated calcium channels and triggers the rapid sealing of the axon membrane at the injury site [79,80,81,82]. A wavefront of the calcium ions propagates anterograde to reach millimolar concentrations in the soma [83], the amplitude of which correlates with the extent of later nerve regeneration [84]. Calcium is also raised throughout the stump by additional calcium entry that follows the reversal of the sodium–calcium exchange pump by the calcium load [80]. Local protein translation proceeds, long-range retrograde signaling is activated, and the local cytoskeleton contributes to the formation of the growth cone [85,86]. The growth cone is composed primarily of a microtubule cytoskeleton and F-actin with the central domain containing microtubules, organelles, and vesicles, and the P domain, composed of dynamic microtubules and F-actin [87]. The F-actin bundles comprise the finger-like projections of the filopodia. 

The axonal calcium activates the nucleotide cAMP, via the calcium-dependent adenylyl cyclase enzyme. In turn, cAMP activates the pro-regenerative kinase DLK via PKA and the transcription factor *CREB1*, amongst others [84,88]. DLK is a key sensor of local injury that informs the soma about the injury [88,89]. JNK downstream of DLK signaling, is also transported to the soma where it activates *STAT3* and *c-Jun* (an immediate early gene of the *AP-1* transcription complex of *Jun*, *c-Fos*, and the *ATF/CREB* families [83]), to promote nerve regeneration [90,91,92]. Accumulating calcium ions activate local intracellular calpain and oxidative species, resulting in axon swelling, rapid granular disintegration of the axonal cytoskeleton [93,94,95,96], and dieback to the first node of Ranvier [97,98,99]. The local calpain at the sealed end of the stump cleaves the submembranous spectrin complex, restructures the cytoskeleton by microtubule and actin depolymerization, and, in turn, allows the elaboration of the growth cone [82,83,100].

The local calcium also activates many other signaling pathways. These include CaMK that phosphorylates nuclear *CREB* which, in turn, later influences gene expression directly. It does so by mediating cAMP-induced transcription [101] via PKA and MEK/Erk pathways [102], and the translation of the transcription factor, HIF-1a that, in turn, activates HIF-1a-responsive genes for axonal regeneration [103]. 

The initial and local translational burst of mTOR controls ~250 localized axonal mRNAs. They transcribe proteins that are retrogradely transported to the soma, facilitated by the local increase in tyrosinated α-tubulin in the microtubular cytoskeleton [104]. The proteins include importin-β1, the adaptor protein that transports cytoplasmic proteins via dynein; the retrograde motor on the microtubules [105], vimentin, STAT3, ZBP-1, RanBP1, Ran being the Ras-related nuclear protein ligand-activated nuclear receptor; and PPARƔ [67,77,105,106,107,108]. *Luman/ATF3*, an ER transmembrane basic leucine zipper transcription factor, is also transported in an importin-dependent manner, to the soma where it is a critical regulator of sensory axon regeneration, linking the unfolded protein response and the ensuing endoplasmic stress response to axon repair [109,110]. The temporal phases of the luman protein levels are coordinated with the three phases of the growth and stress responses immediately after injury, the pre-regenerative phase, 9 to 24 h thereafter when transcription factor activity regulates DNA replication and transcription, and the regenerative phase at 4 days [111].

The transported proteins activate pro-regenerative pathways in the second slow signaling phase of nerve injury, including translation and activation of transcription factors, specific epigenetic modifiers, and additional signaling molecules [66,67,112].

There are >1500 transcription factors in the genome [113] of which *c-Jun* is markedly increased in axotomized motor and sensory neurons [114]. Transcription factors such as *AP1*, *CREB*, and *ATF3* serve as ‘hubs’, those genes with many connections that coordinate the activity of connected genes [115,116,117]. Transcription factors bind to selective DNA promotor regions to increase or repress specific target gene transcription to coordinate multiple RAG expression [67,116,117,118,119]. The transcription factor *c-Jun* that coordinates the transcription of many RAGs was the first to be identified in a RAG network [114,120]. CREB protein regulates the transcription of *BDNF* and *arginase 1*, amongst other genes, and drives the transcription of both the *AP1* and the *ATF3* hub genes [117]. One study distilled the RAG network to ~40 transcription factors downstream of multiple parallel signaling pathways [121], of which the calcium-dependent cAMP activates only a fraction of injury-induced genes, at least in sensory neurons [122].

The independently identified transcription factors include *ATF3* [123,124,125,126], *STAT3* [127], *SOX11* [128,129,130,131], *SMAD1* [132,133,134,135], *C/EBPβ* [136], *p53* [137,138], and *KLF4* [139,140]. The cAMP-dependent transcription factor *ATF-3* is upregulated rapidly in injured DRGs and motoneurons following JNK signaling [76,123,124,125,126] and is used frequently and reliably as a biomarker of axotomy [23,76,113,126]. Both *Jun* and *ATF3* mediate PNS regeneration *in vitro* [125] and *in vivo* [123,126]. Phosphorylated *STAT3* stimulates growth initiation but does not perpetuate axonal growth [127]. *SOX11* is also elevated in injured nerves and promotes their regeneration [128,129,130]. It does so via the activation of *ATF3* and *c-Jun*, and the RAGs, *Arpc3* and *Sprr1a* [129], and by increasing the responsiveness of neurotrophic factors [130]. Elevated *SMAD*s are phosphorylated and accumulate in the nuclei of injured DRGs and motoneurons, *SMAD1* in the sensory neurons [131,132,133,134], and *SMADs 1,2*, and *SMAD*4 in the motoneurons [135]. They act as modulators of activated *GSK3β* downstream of the P13K/Akt pathway, by interacting with the transcriptional coregulator *p300 HAT* to promote the expression of several pro-regenerative target genes and, in turn, nerve regeneration [132,133,135]. *C/EBPβ* that is induced in injured neurons binds to the promoters of *Tα1-tubulin* of the microtubules and the growth cone protein *GAP-43*, as an essential transcription factor for the injury response [136]. The transcription factor *p53* stimulates the regeneration of transected sciatic nerve fibres, and in p53 knockout mice, the muscle reinnervation by injured facial nerves was reduced [137,138]. In contrast to the other factors, *KLF4* is downregulated after injury and thereby negates its inhibitory effect on nerve regeneration [139,140].

Epigenetic modifications, or “tags”, regulate patterns of gene expression by altering DNA accessibility and chromatin structure without altering the DNA sequence [141]. They include miRNA (microRNA) and the non-coding RNAs, namely, long-chain and circular RNAs. They can affect nerve regeneration by altering transcription factor access to DNA by methylating DNA, post-translational modification of the histone protein that wraps around nuclear DNA, or by controlling ncRNAs (noncoding RNA) that silence genes [142,143].

DNA methylation generally is associated with the repression of transcription [67]. There are >200 identified histone modifications [144] of which (PCAF)-dependent acetylation of *H3K9ac* with reduced *H3K9* methylation, is an important example. Reduced methylation relaxes the chromatin environment surrounding the promoters of several pro-regenerative genes. In sensory neurons, this modification results in the expression of the GAGs, *GAP-43*, *galanin*, and *BDNF* [145]. Acetylation requires retrograde ERK signaling. Also, *PCAF* overexpression promotes axonal regeneration of the central axons of the DRG neurons across injured spinal cord [145].

MicroRNAs (miRNA), of which lncRNA (long-chain non-coding RNAs) account for 60 to 80% of the mammalian genome transcriptome [146], are differentially expressed after peripheral nerve injury [143,147]. They target specific mRNAs with resulting repression or degradation of their translation. For example, *miR-21* is upregulated after sciatic nerve injury and, by targeting *Sprouty2* (a specific inhibitor of the Ras/Raf/Erk pathway), promotes axonal growth from adult DRG neurons [148,149]. A second example is the nerve regeneration that results from miR-26a that specifically targets *GSK3β* to rescue regeneration, the miR26a-GSK3β pathway regulating regeneration at the neuronal soma by controlling the expression of the regeneration-associated transcription factor, *SMAD1* [149].

### 2.3. Wallerian Degeneration

The isolation of the peripheral nerve fibres distal to sites of crush or transection from their neuronal cell bodies deprives them, for all intents and purposes, from their source of protein, lipid, glycoprotein, and carbohydrate synthesis, leading to the self-destructive process of Wallerian degeneration [51,63,94]. The axons in the proximal nerve stump die back to the first node of Ranvier, preventing the scarring that occurs in CNS injury [97,98,99,150]. The calcium ions entering the nerves, activate calpain proximal and distal to the injury site to mediate proteolysis with degeneration of axon segments several hundred micrometers from the site [151,152]. When the axonal transport of NMNAT2 is interrupted by calcium-dependent proteolysis of the cytoskeleton, the axonal NMNAT2 degrades and NMN accumulates in the nerve stump. The NMN elevates the normally controlled *SARM1* expression with the result that the SARM1 protein that is essential for axon degeneration, rises [153]. The downstream steps to axon degeneration remain to be determined. Meanwhile, the remaining fast axonal transport allows for continued propagation of action potentials in the distal stump for hours and even days [93,99,154,155,156,157].

Ras/raf/ERK signaling in SCs is evident immediately after injury with ERK levels returning to lower levels just prior to SC proliferation [158]. Raf-kinase activation drives SC dedifferentiation as well as inducing much of the inflammatory response that is important for nerve repair, including breakdown of the blood–nerve barrier and delayed macrophage recruitment into the denervated nerve stump [159]. The myelin sheaths constituting ~50% of the total myelin [160], are broken down by SC autophagy [161,162,163]. The SC expression of proinflammatory cytokines and chemokines within 3 to 5 h of nerve injury contributes to myelin and axon breakdown and the phagocytosis of their debris within the first 3 days of injury [164]. The cytokines, including IL-1α and LIF, their receptors IL-6R and gp130, respectively, and TNF-α, stimulate the expression of the two cystolic forms of PLA2 [165]. These remain high for two weeks [165]. TNF-α hydrolyses the phosphatidylcholines in the myelin membranes, releasing the potent myelinolytic agent, lysophosphotidylcholine. Lysophosphotidylcholine also feeds back to sustain cytokine expression and the expression of the chemokines, including MIP-1, MMP-1α or CCL2, and IL-lβ [52,166,167,168,169,170,171]. Within a day of the nerve injury, IL-6 and TNF-α induce the expression of MMP-9 that contributes to myelinolysis [172,173].

The recruited macrophages also express these cytokines and chemokines and are responsible for the bulk of the myelin and axonal phagocytosis over the following ~3 weeks [99,174]. They play a major role in removing inhibitory molecules, including MAGs, from the degenerating axons [175]. Their release of nitric oxide has been implicated in myelin breakdown [176]. Their phagocytosis of myelin debris includes the myelin that is opsonized by complement components binding to the complement receptor type 3 on the macrophages [177]. Antibodies to non-opsonized myelin are phagocytosed via the macrophage Fc receptor [178]. The third macrophage receptor used in phagocytosis is the scavenger receptor-AI,II [179]. The macrophage spectrum was separated into two “polarizing” phenotypes, M1 and M2, with the M1 macrophages associated with pro-inflammatory and neurodegenerative functions and the M2 macrophages broadly viewed as anti-inflammatory and promoting cellular repair [180]. There is an early increase in M1 macrophages 1–2 days after injury, with M2 macrophages replacing these from days 3 to 7. The majority of the accumulating M2 macrophages may be of a mixed phenotype because these mixed-type macrophages were not included in their analysis [180,181]. In addition to their essential role in axonal and myelin phagocytosis, the macrophages sense hypoxic conditions and stimulate angiogenesis for a polarized vasculature that guides SCs and elongating axons [182,183].

### 2.4. Schwann Cell Response to Nerve Injury

Once SCs lose their myelin, they re-enter the cell cycle, proliferate by mitosis, transition from their myelinating to a growth state to support nerve regeneration across an injury site, and into the denervated nerve stump [51,56,127,184]. The transient SC expression of Cdc2, possibly induced by *c-Jun* [185], is involved in their proliferation and migration [186]. The SC genes that transcribe myelin proteins, including *MAP*, *MBP*, *P*_0_, and *PLP*, are downregulated in the denervated distal stump, while RAGs associated with nerve regeneration, are upregulated as the SCs dedifferentiate and acquire the ability to survive without axonal interactions (Figure 1A; [187]). The SCs are programmed by *c-Jun* to generate repair cells that are essential for nerve regeneration, with *c-Jun* accelerating the downregulation of myelin genes, promoting myelin breakdown, and amplifying the upregulation of a broad spectrum of repair-supportive features, including the expression of trophic factors [188,189,190]. As early as 1991, the upregulation of hundreds of growth-supportive RAGs was reported in denervated SCs [191] with more genes differentially expressed in the SCs than in sensory neurons [192].

IL-6, synthesized in the SCs within 24 h [166,168,193,194], signals the expression of RAGs via its receptor [167]. The SCs express NRG-1, a member of the family of glial growth factors, and its ErbB2/3 receptor [195,196,197,198]. NRG-1 levels remain elevated for at least 30 days [199,200,201]. NRG-1 strongly inhibits the expression of genes involved in myelination and glial cell differentiation, suggesting that it might be involved in the SC dedifferentiation from the myelinating to the repair phenotype [202]. In addition, NRG-1 likely mediates, at least in part, the second phase of SC proliferation that is stimulated when regenerating axons contact the SCs in the Bands of Büngner [203,204,205]. The scaffolding oncoprotein Gab2, is required for SC proliferation after nerve injury, its activation leading to SC migration, possibly through actin modulation [206,207]. In a model based on their findings, SC migration is promoted by autocrine/paracrine activation by NRG of its SC erbB2 receptor that results in transcriptional *Gab2* expression via the Rac-JNK-cJun pathway and *GAB2* phosphorylation via the paracrine HGF from fibroblasts [206]. Notch, a transmembrane protein that is also upregulated in denervated SCs, promotes their proliferation and is downregulated as myelination proceeds [208].

The non-coding RNAs, namely the long-chain and circular RNAs, play important roles in SC proliferation and migration [143]. Examples of the long-chain RNAs include NEAT1 that promotes SC proliferation and migration [143], MALAT1 that elevates BDNF [209], and Loc680254 that promotes nerve regeneration by inducing SC proliferation [210]. BC088259, which showed the most significant upregulation after sciatic nerve injury, interacts with vimentin to regulate SC migration [211,212]. Downregulation of some long-chain RNAs also enhances SC proliferation and migration. An example is MEG-3 that increases SC proliferation and migration and facilitates nerve regeneration through the PTEN/P13K/ADT pathway [213]. Some circular RNAs are also upregulated in denervated nerve stumps and are associated with SC proliferation. For example, cirRNA-Spidr targets P13K-Akt to promote nerve regeneration after rat sciatic nerve crush injury [214].

Denervated SCs express several neurotrophic factors and their receptors within 7 days, their levels peaking within a month. These factors include neurotrophins, *NGF*, and *BDNF* with their p75 receptor *p75NTR*, *NT-3*, *NT-4/5* and the *trkC* receptor, the *GDNF* family and their receptors *GDFRα1* and *ret*, and other factors including *IGF-1* and *IGF-II*, *VEGF*, *HGF*, *PDGF-BB*, *FGF*, *TFG-β* and their receptors, and *PTN* [52,215,216,217,218,219,220]. The expression of *GDNF* and *PTN* is specific for the denervated SCs in the motor pathways of the quadriceps nerve branch of the femoral nerve, whereas the remaining neurotrophic factors, *HGF*, *BDNF*, *NGF*, and *IGF-1* and *IGF-II*, are more specific for the denervated SCs that are located in the sensory pathways of the saphenous nerve branch [216,217,218]. The time course of expression varies for different trophic factors. *NGF* rises rapidly and then declines prior to a 5-fold upregulation, possibly in response to macrophage release of IL-1β, persisting for at least 3 weeks [221,222]. Upregulation of *BDNF* is much slower, being detectable at 7 days after nerve injury and increasing up to 28 days to levels much higher than those of *NGF* [52,221]. *GDNF* and its *GDFRα1*, but not its coreceptor *Ret*, are upregulated more quickly, reaching a peak within 7 days [223]. The expression is not sustained, declining with time when the distal nerve stump denervation is prolonged for >1 month ([52,224] see Section 3.2). NT-3 upregulates SC *c-Jun* [189] and regulates the SC levels of *p75NTR* [118,119,188].

### 2.5. Axonal Regeneration

Regenerating axonal sprouts emanate from the first node of Ranvier proximal to the injury site [1,225,226]. The growth cones form without direct support from the cell body and depend on axonal material in the proximal stump, including the preexisting cytoskeletal elements of actin and tubulin [226,227]. Anterograde axonal transport delivers most of the materials for subsequent axonal elongation [70]. The growth cones link to the organized ECM glycoproteins at the injury site, navigate across the site, and regenerate axons along the repair SC layer of the Bands of Büngner on the endothelial laminal sheath in the denervated distal stumps [187]. The growing axons contact and interact with the SC basement membrane glycoproteins. These include collagen, fibronectin, tenascin C, and laminin, that are secreted by SCs and fibroblasts [228,229,230,231,232] as they progress through the distal nerve stump via SC adaptor molecules such as N-cadherin and integrins [233,234,235,236,237].

A single regenerating axon can give rise to as many as 50–100 branches [238] but it is an average of five daughter axons that regenerate, more of which regenerate into the distal stump after crush than after transection injuries [239,240]. The remyelination of regenerating axons is initiated by their contact with SCs that form myelin layers (lamellae) in proportion to the size of the regenerating axons [241]. Both axonal NRG-III and SC NRG-1 play pivotal roles in remyelination [200,242] with tyrosine phosphorylation of SC GAB1, principally regulated by NRG-1, being essential for remyelination [206]. The internodal distances between the reformed myelin sheaths are shorter than normal because of the 3-fold increase in SC numbers after denervation [243] but they do lengthen with time [241]. The diameter of the regenerating axons increases in proportion to the size of their parent nerve [244], recovering their original size if and when they make functional connections [72]. The way the excess axonal branches are removed with time is not well understood.

## 3. Poor Recovery of Function after Peripheral Nerve Injury and Repair 

The timing of surgical intervention after nerve transection depends on whether the transection is ragged or sharp [245,246,247,248]. Sharp injuries from knives or razors should be repaired within 3 days but the repair of ragged transections from blast, gunshot, fracture, or crush injuries is usually delayed for at least 3 weeks to allow the nerve ends to be demarcated [248]. Frequently, an early surgical evaluation is made of whether any of the nerve remains in continuity. When the nerve is transected, the nerve stumps may be sutured to a nearby local structure to prevent their retraction and to facilitate later surgical repair. Especially when the nerve injury is associated with comorbidities that include vascular problems, the later repair is undertaken once the local inflammation has subsided [249,250]. Irrespective of whether or not surgical repairs are delayed, proximal injuries in particular, result in long delays before the regenerating nerve fibres contact distal denervated targets (Figure 1B). Even when proximal injuries that include brachial plexus nerve injuries, undergo early repair, many of the axotomized neurons remain without target contacts for periods of two or more years (chronic axotomy) as they regenerate their axons slowly over long distances at a regeneration rate of 1 mm/day. The rate was determined with the Tinel sign that identifies the site at which a tap on the regenerating nerve elicits a tingling sensation in a conscious patient [249,250]. The SCs in the denervated nerve stumps are subjected progressively to chronic denervation, especially those far distal to the microsurgical suture of the proximal and distal nerve stumps. The accepted explanation for the observed poor recovery of function after proximal nerve injuries is that the denervated targets deteriorate with massive muscle atrophy and, in turn, fat replacement of denervated muscle [2].

### 3.1. Time-Related Decline in Nerve Regenerative Capacity

A cross-suture technique was pioneered by Holmes and Young in 1942 to examine their hypothesis that chronic motoneuron axotomy accounts for poor functional recovery after surgical repair [251]. After delaying the suture of the proximal stump of a cut hindlimb nerve to a freshly denervated stump of a second different hindlimb nerve in rabbits, they reported that the denervated muscle recovered normal levels of contractile force [251]. This finding negated their hypothesis, leading them to conclude that chronic muscle denervation rather than chronic axotomy was the basis for the reported poor functional recovery after delayed surgical repair or delayed reinnervation of denervated muscles. This conclusion has remained the accepted view ever since. More recently, we systematically examined each of the contributions of chronic axotomy, chronic denervation of the distal nerve stump, and chronic denervation of target musculature, to poor recovery after nerve transection (Figure 2; [24,62,252,253,254,255,256,257,258,259]). Our surgical technique was to prolong TIB motoneuronal axotomy or the denervation of the CP distal nerve stump prior to the cross-suture of TIB and CP proximal and distal nerve stumps, respectively (Figure 2A,B). After at least 5 months, we determined how many motoneurons (1) reinnervated TA muscle fibres with motor unit number estimation (MUNE), the ratio of the whole muscle to mean MU isometric contractile forces (Figure 2C), and (2) regenerated their nerve fibers into the CP nerve stump by retrograde dye labelling of the neurons with either of two fluorescent dyes, FG or FR. (Figure 2D–F).

Contrary to early conclusions [251], chronic motoneuron axotomy had a pronounced negative effect on the regeneration of nerve fibres: both the numbers of (MN) motoneurons that regenerated their axons in the freshly denervated nerve stumps and the numbers of MUs, the motoneurons whose nerves reinnervated muscle fibres, declined exponentially with a time constant (Ƭ) of 40 days, to reach a plateau of 33% of the numbers after immediate nerve repair (Figure 3A,C; [62]). It was the capacity of regenerating motor nerve fibres to reinnervate 3–5 times as many denervated fibres as they normally do [62], the same capacity as that of intact nerve fibres sprouting in partially denervated and reinnervated muscles [260,261,262,263], which resulted in the reinnervation of all the denervated muscle fibres and, with their full recovery from muscle atrophy, the full recovery of the reinnervated muscle contractile force [62]. It was the same full recovery of muscle force after reinnervation by chronically axotomized motoneurons that was the basis for the faulty conclusion made previously in 1942 [251]. It is also the reason why the clinical assessments of reinnervated skeletal muscle strength by the MRC 5-point scale [264] or by revised scales [265] are frequently misleading. These clinical assessments continue to underestimate the deleterious effects of chronic axotomy on nerve regeneration and muscle reinnervation. It is interesting, as an aside, that the size of the nerve fibres that supply the enlarged MUs are not enlarged [266] and the regenerated nerve fibres recover their normal size once they make functional connection with denervated muscle fibres [62].

Chronic denervation of the distal nerve stump and the muscles that the intact nerve normally supply, is even more deleterious to nerve regeneration than chronic motoneuron axotomy (Figure 3B,D; [251,253,257,258,259]). The exponential decline in numbers of MUs and freshly axotomized motoneurons that regenerate their axons into the chronically denervated nerve stumps plateaued at ~5% at 180 days of chronic denervation (Figure 3B; [252]). To determine whether poor regeneration was due to chronic SC denervation within the distal nerve stump and/or prolonged muscle denervation, a chronically denervated autograft was inserted between a freshly cut TIB proximal nerve stump and a chronically denervated CP distal nerve stump [266]. The experiments demonstrated that it is the chronic denervation of the SCs and not the chronic denervation of the TA muscle that was the causative factor of poor nerve regeneration through chronically denervated distal nerve stumps [259]. Whilst the SC basement membranes are disrupted, their columns shrink [267,268,269,270,271]. As many as 90% of rat SCs die after 17 months of chronic denervation (Figure 4A–C). The chronically denervated SCs proliferate in response to mitogens and they support myelination *in vitro* ((Figure 4C–E) [267]) and *in vivo* [272], and support the full recovery of regenerated nerve fibres *in vivo* (Figure 4F–H) after 6–25 months of sciatic distal nerve stumps and intramuscular nerve stumps in rats and rabbits [272,273]. STAT3 has been implicated in the long-term maintenance of SCs [274], and *c-Jun* influences both apoptosis and proliferation of denervated SCs [275,276,277].

The small number of nerve fibres regenerating through chronically denervated nerve stumps reinnervated three times as many TA muscle fibres as normal but was not sufficient to reinnervate many of the denervated muscle fibres [251]. In addition, the reinnervated muscle fibres failed to recover their normal size from their denervation atrophy [252]. The atrophy proceeds as the size of the denervated fibre declines rapidly within the first two weeks and falls more gradually thereafter to 2% of its normal size two years after chronic denervation [278]. There is also progressive mitochondrial dysfunction with accompanying upregulation of the miRNA miR142a-5p and subsequent lysosomal degradation [279]. Nonetheless, the surviving denervated fibres conserve their fascicular arrangement and their functionally important proteins, including acetylcholine receptors and N-CAMs [279,280,281].

We had suggested that the failure of reinnervated muscle fibres to regain their former size after the chronic denervation that proceeded nerve repair resulted from declining numbers of satellite cells to provide nuclei to reinnervated fibres [252]. The satellite cells are a population of heterogenous stem cells that are normally mitotically quiescent and express Pax7 [282,283,284]. They lie between the sarcolemma of the myofibres and the ensheathing basal lamina [285] and are activated upon nerve and/or muscle injury [284]. They divide by asymmetrical division to produce myogenic progenitor cells, and by symmetrical division to produce multiple satellite cells that contribute nuclei to atrophic denervated muscle fibres as they recover their size after reinnervation [282,283,284]. The activated satellite cells differentiate into myocytes and co-express *Pax7* and the muscle-specific basic helix–loop–helix protein, *MyD*, of the *MRF* family of transcription factors [282,283,284,285]. The number of *MyD*-positive cells was reported to decline in chronically denervated biceps muscle of patients as a function of the time to surgical repair of their brachial plexus injury [286]. Examination of their data, however, indicated that the data were more readily fitted by an exponential rather than the reported linear decline. The *MyD*-expressing cells declined to a low plateau rather than to zero. Earlier morphological studies of chronically denervated frog muscles had also suggested that the proliferative capacity of the satellite cell pool is exhaustible [287], and evidence of repeated cycles of muscle fibre degeneration and regeneration [288] indicated but did not prove that the ultimate fate of chronically denervated muscle fibres is their replacement by fatty and connective tissues [289]. The question remains as to whether the explanation for incomplete recovery of reinnervated muscle fibre diameters after chronic denervation is that the activated satellite population in the denervated muscles is depleted with time and/or that the cells fail to divide and express *MyD*.

### 3.2. Transient Expression of Regeneration-Associated Genes

Addressing why neuronal regenerative capacity and SC growth support decline with time and distance, we reasoned that elevated RAG expression by axotomized motoneurons and/or denervated SCs declines with time after chronic nerve injury, a hypothesis that proved to be correct [224,290,291,292]. The parallel decline in RAG expression, which likely accounts, at least in part, for poor functional recovery, is illustrated by the exponential decline of the expression and translation of *tubulin*, *actin*, and *GAP-43* in chronically axotomized motoneurons [291] (Figure 5): *p75* and p75 [290,291,293] (Figure 6 and Figure 7), *c-Jun* [293] (Figure 6), and *GNDF* [224] (Figure 7) in chronically denervated SCs. The elevated mRNA levels of the growth-associated proteins in response to a second nerve section, a refreshment injury 1–6 months after the first axotomy (Figure 5F), decayed more rapidly than after the first axotomy (Figure 5H). Similar findings were reported for facial motoneurons [292]. Downstream of *c-Jun*, the expression of *GDNF* specifically, and not other proteins that include *NTN*, *PSP*, and *ART*, declines exponentially in the chronically denervated nerve stumps (Figure 7E), the latter genes being reduced in *c-Jun*-deficient mice [119,188,189]. The transient nature of gene expression of neurotrophins and their receptors in motoneurons is summarized in Figure 2 of an extensive review [52].

The *Shh* gene that is not expressed in either developing SCs or in intact nerves, is upregulated strongly in SCs after injury [294,295] and improves nerve regeneration in several settings [296,297,298,299,300,301]. Applied Shh elevates *c-Jun* in cultured SCs and the expression of both *Shh* and *c-Jun* declined *in vivo* during chronic denervation [293]. Furthermore, *c-Jun* activation and expression of its target *p75NTR* was reduced by conditional knockout of the SC *Shh* gene. Inhibiting Shh signaling also reduces SC *BDNF* expression, motoneuron survival after injury, and axon regeneration [294,295].

The identified *c-Jun* gene-regulated set of *Aqp5*, *Gpr37L1*, *Igfbp2*, and *Olig1* is highly enriched amongst the genes affected by chronic denervation, and it correlates with both *c-Jun* levels and regeneration [293]. *Igfbp2*, for example, promotes Akt phosphorylation, a pathway linked to SC proliferation and differentiation [190,199,298,299]. Gpr37L1 is a receptor for prosaposin and prosapeptide [301] that are secreted after nerve injury and facilitate regeneration [302]. In experiments on chronic denervation, this gene group encompasses *Cxcl5*, *Egfl8*, *Gas2I3*, *Megf10*, and *Pcdh20*, all of which are upregulated in SCs after injury [163,303,304,305]. *Cxcl5* activates *STAT3* [304], the transcription factor that is important for maintaining repair cells during chronic denervation [274]. *Gas2I3* has a role in the cell cycle, and *Megf10* in phagocytosis [306,307].

## 4. Exogenous and Endogenous Neurotrophic Factors

### 4.1. Exogenous Application of Neurotrophic Factors

The demonstration that NGF evoked dramatic neurite outgrowth from sympathetic and DRG (dorsal root ganglion) neurons NGF [308]) generated many *in vitro* and *in vivo* studies which advocated that growth factors, including BDNF and GDNF, promote nerve regeneration [52,53,59]. These studies used several outcomes measures of nerve regeneration to evaluate the effects of exogenous neurotrophic factors, including axon counts distal to the site of injury, the ‘pinch test’ that evaluates sensory nerve regeneration in animals by identifying the most distal point of the regenerating nerves at which a nerve pinch with fine forceps elicits an intake of breath in the lightly anesthetized animal [309,310,311,312,313,314,315], and walking track analysis to evaluate motor nerve regeneration [316]. While having some clinical relevance, the relevance of these measures for nerve regeneration is negated by their failure to consider the outgrowth of several axonal sprouts from the proximal stump of injured nerves in the PNS [51,256,263] and CNS [317]. Thereby, the positive effects of neurotrophic factors on how many axotomized neurons (1) regenerate their axons, (2) reinnervate denervated targets, and (3) result in functional recovery, have been overestimated. Hence, we adopted retrograde labelling of neurons to examine how many regenerated their axons into the distal nerve stump in response to exogenous application of neurotrophic factors in addition to counting the regenerated axons in the distal stump.

BDNF and/or GDNF Retrograde labelling revealed that the reduced numbers of chronically axotomized TIB motoneurons regenerating their axons into a freshly denervated CP distal nerve stump were elevated significantly by the exogenous growth factors (Figure 8; [256]). BDNF was effective only in low doses and not at high doses due to the binding of BDNF to both trkB and p75NTR receptors on the motor nerve membranes, trkB and p75NTR, mediating positive and negative effects on nerve regeneration, respectively [255,256]. GDNF on the other hand, was effective at all doses, acting via ret and GFR-α1 receptors [256]. Whilst GDNF administration to the cross-sutured TIB–CP nerves promoted nerve regeneration after delayed but not immediate repair [256], a 2- and/or 4-week delivery of GDNF encased in polymeric microspheres, to the suture sites of a 10 mm long ANA (acellular nerve graft) placed between CP nerve stumps (Figure 8D; [318]), was effective in elevating the numbers of CP motor and sensory neurons that regenerated their axons to normal levels (Figure 8E,F; [318]). Unfortunately, though, the regeneration enhancing effects of GDNF and BDNF were confounded by their propensity to promote outgrowth and regeneration of multiple axonal sprouts after nerve repair [256]. This was the reason why we considered increasing endogenous rather than exogenous neurotrophic factors to promote nerve regeneration.

### 4.2. Endogenous Neurotrophic Factors

A striking accelerated and amplified expression of *BDNF* and its *trkB* receptor in axotomized neurons that was evoked by a 1 h period of low-frequency (20 Hz) electrical stimulation (ES) was followed by accelerated and amplified expression of *tubulin* and an accompanying reduction in *neurofilament* expression (Figure 9 [75]). The time course of the gene expression suggests causation between neurotrophic factor upregulation, RAG expression, and accelerated nerve regeneration, as discussed in Section 5.

## 5. Neuronal Activity and Nerve Regeneration

### 5.1. Reduced Activity and Synapse Withdrawal

We addressed the question of whether the decline in regenerative capacity of chronically axotomized neurons over time/distance is due to reduced neuronal activity and, in turn, to their reduced interaction with denervated SCs in the distal nerve stump. Cross-correlation techniques to examine cat hindlimb neural activity during treadmill walking before and after nerve injury with and without surgical repair revealed a fall in motor and sensory nerve activity that results from synaptic withdrawal from motoneurons and by the loss of contact of the sensory nerves with their denervated sense organs, respectively (Figure 10A,B; [319]). The reduced motor activity and demonstrated synaptic depression in axotomized motoneurons [319,320,321] accounted for the decline in the excitatory VGLUT1-positive glutamatergic and inhibitory GAD67-positive GABAergic synaptic boutons on the motoneurons [322,323,324,325,326,327] (Figure 10C–E; [58]) that were linked to astrocytic activation [328]. Synaptic loss was reversed by daily treadmill exercise implemented 3 days after nerve transection and surgical repair but not later, the reversal depending on slow, continuous exercise in males and interval training exercise (short high-speed sprints followed by rest periods) in female mice [329,330] (Figure 10D,E; [58]). The efficacy of exercise is likely BDNF dependent because synapses withdraw from intact motoneurons in conditional BDNF knockout transgenic mice [331]. These findings (1) elucidate the profound reduction in neural activity that results from the withdrawal of synapses from axotomized motoneurons and the loss of sensory neural activity after injury, and (2). describe how appropriate patterned treadmill exercise prevents withdrawal in a BDNF-dependent manner.

### 5.2. Staggered Nerve Regeneration

The rat femoral nerve, introduced as a model to study peripheral nerve regeneration [332], was used later for counting retrogradely labeled motoneurons that had regenerated their axons to demonstrate (1) ‘staggering’ of regenerating axons across the site of femoral nerve transection and microsurgical repair (Figure 11); [20]), and (2) preferential reinnervation of the quadriceps motor branch by motor nerves and of the cutaneous sensory branch by sensory nerves (Figure 12A; [20,22,333]). The staggering was evident by the lengthy 8–10-week period for all the motoneurons to regenerate their axons to the 25 mm distance to fluorescent dye application (Figure 12A), a much longer time period than expected from the 3 mm/day regeneration rate [334] and the 4 week period for all these motoneurons to regenerate axons across the femoral nerve suture site (Figure 11F; [21]). There was some delay attributed to the 10-day period for the longitudinal alignment of laminin and SCs across the suture site [335].

### 5.3. Preferential Reinnervation and Growth Factors

The preferential upregulation of *PTN* and *GDNF* in denervated SCs (Figure 13C; [216,218]) accounts for the preferential growth of regenerating femoral motor nerve fibres in their appropriate motor nerve branch (Figure 13A,D (D: green and purple dotted lines); [20,336]). *PTN* and *GNDF* expression increases 4 days after denervation (Figure 13C) when few motoneurons regenerate axons across the surgical repair site (Figure 13B) and randomly reinnervate appropriate and inappropriate branches at 2 and 3 weeks (Figure 13D, blue and red dotted line).). By 14 days, when the growth factor expression is maximal in motor SCs (Figure 13C), ~70% of the femoral motoneurons have regenerated axons across the suture line (Figure 12A). By 4 weeks, the axons grow preferentially into the endoneurial tubes that previously surrounded motor nerve fibres and continue to do so with the growth factors remaining (Figure 13C,D, D: green and purple dotted lines). There is a similar explanation for the preferential reinnervation of sensory pathways by sensory neurons [336]. Studies of the growth of motor axons into femoral nerve branches in *NCAM*-/- knockout mice demonstrated that N-CAM is also required for preferential reinnervation of the motor branch with motor axons expressing polysialylated *NCAM* that reduces axon–axon adhesion [337]. 

### 5.4. Low-Frequency Electrical Stimulation Accelerates Axon Outgrowth

That continuous 20 Hz ES and a 15–60 min 20 Hz ES of a crushed nerve accelerated muscle reinnervation [18] and the plantar extensor reflex [19], respectively, led us to determine whether ES accelerates nerve regeneration itself and not the formation of functional neuromuscular connections. We delivered bipolar continuous electrical pulses at a 2x-threshold voltage and 20 Hz frequency to the femoral nerve proximal to the site of femoral nerve transection and microsurgical repair [20]. Continuous and shortened periods of 1 to 3 h 20 Hz ES dramatically shortened the period for all the motoneurons to regenerate their axons across the suture site and into the motor and sensory nerve branches, maintaining the preferential reinnervation of the motor branch by regenerating motor axons (Figure 12A,B; [20]). The ES effect was mediated by action potentials conducted to the neuronal soma, the effect being eliminated by preventing the action potential conduction to the soma with a TTX block central to the ES site (Figure 12C,D; [20,23]). The sensory neurons did not show a preference for the cutaneous sensory branch of the femoral nerve in the experiments of Geremia et al. [23], but significant preference was demonstrated by those of Brushart et al. [22]. In addition to impulse conduction, *BDNF*, which is upregulated by ES [75], is essential, the ES-induced enhancement of the specific regeneration of femoral motor axons into the motor branch blocked by an infusion of a BDNF antibody (Figure 14). That ES did not increase the rate of slow axonal transport, and hence, regeneration rate, the efficacy of ES in promoting nerve regeneration was attributed to accelerated axonal outgrowth across the suture site and not to an increase in the rate of regeneration ES [21] in contrast with the conditioning lesion (CL) that accelerates axonal outgrowth and regeneration rate (Section 5.7).

The efficacy of ES in promoting nerve regeneration [20,21,22,23,24] has been confirmed [18,19,26,27,28,29,30,31,32,33,34,35,36,37,38,39,40,41,42,43] and reviewed [53,56,57,58,59,60,61]. There are additional studies demonstrating that ES (1) accelerates nerve regeneration through a nerve isograft of 10 mm in *Lewis* rats [46] and through 10 and 20 mm nerve autografts in *Sprague-Dawley* rats [47], (2) increases the number of macrophages and neutrophils in the denervated distal nerve stumps as well as the number of M2 macrophages within autografts [46,50], (3) shifts the macrophage phenotype in a locally demyelinated peripheral nerve from the proinflammatory M1 toward the predominantly pro-repair M2 type [338], (4) accelerates Wallerian degeneration and upregulates BDNF and NGF in denervated SCs [339], and (5) together with testosterone administration promotes facial nerve regeneration and functional recovery of whisking and blink reflexes after crush injury in castrated male rats [33,36] in association with a more rapid and sustained upregulation of *BDNF* than either alone [36]. 

The efficacy of daily 1.5 h periods of ES over 6–8 weeks in promoting muscle reinnervation and functional recovery after direct nerve repair [48,340] or via an isograft [49] was the same as that after a 1 h or 2-week period of ES [20,32]. The positive effects of transcutaneous ES [341] and 20 min nerve ES after delayed nerve repair [44] were as low as 10% of the effect of the 1 h ES [20]. Also, a reported effective 10 min ES [342,343,344] was not supported by any significant recovery after a sciatic nerve isograft repair in a later study [345].

### 5.5. ES Promotes Axon Outgrowth after Delayed Surgery

ES also reverses the regression of the regenerative capacity of chronically injured nerves (Figure 15; [24]). After ligating either the CP proximal nerve stump and/or the distal TIB stump for two months prior to cross-suture repair, ES elevated the numbers of motor and sensory neurons that regenerated their axons to the same levels as after immediate cross-suture (Figure 15B,C; [24]). The ES increased both the number of regenerated nerves and those that reinnervated muscle (Figure 15B,C; the lowest two histograms), demonstrating that ES of chronically injured nerves accelerated both their regeneration and their target reinnervation. Neuronal excitation with bioluminescent optogenetics also accelerated nerve regeneration and muscle reinnervation after immediate and delayed nerve repair [346].

### 5.6. Exercise and Axonal Regeneration

The rationale for examining whether exercise like ES promotes nerve regeneration was the ES-induced accelerated rise in *BDNF*, *trkB*, and *NT4/5* expression in axotomized motoneurons (Figure 9D,E; [28,32,75]). Many studies using several different exercise protocols reported enhanced sensory and motor nerve regeneration [25,32,347,348,349,350,351,352,353,354,355,356,357,358,359]. Daily exercise, like ES, accelerates muscle reinnervation by regenerating nerves, with combined exercise and ES being the most effective paradigm in the early phase of regeneration [32]. Other studies reported significantly greater axon outgrowth after a 2-week period of moderate daily exercise was initiated 3 days after surgical repair of a transected nerve, as compared to immediate ES [353,354,358].

That androgen receptor signaling regulates the expression of *BDNF* mRNA [360] spurred investigations of the effects of daily testosterone administration with and without ES on nerve regeneration [33,36,359]. Either testosterone or ES was found to promote facial nerve regeneration and recovery of whisking [33], the combination of both having the most prolonged and maximum effect [33,36,359]. English and colleagues, who had reported that daily exercise prevented a loss of synaptic contacts on axotomized motoneurons (Figure 10D,E; [329,330]), showed that the androgen receptor antagonist, flutamide, eliminated both the ES and the exercise effects of increasing the length of regenerating axons in mouse hindlimbs [359].

Misdirection and abnormal antagonistic muscle activation are problems even after enhanced nerve regeneration by ES and/or exercise and/or testosterone administration. The problem is particularly severe after large nerve transection injuries. For example, after rat sciatic nerve injury, 71% of regenerating CP motor nerves reinnervated two muscles after crush injury, 42% after autograft repair, and 25% after autograft repair, but recovery of ankle motion and balance was incomplete in all cases [361]. Poor recovery is consistent with the random reinnervation of distal stumps after sciatic nerve transection and microsurgical repair [362] in contrast to the emerging preferential reinnervation of the motor branch of the femoral nerve by regenerating motor nerves (Figure 12A; [20,333,335,363]). ES exacerbates the misdirection of regenerating sciatic nerve fibres [364] with reduced TIB nerve contribution to the innervation of the triceps surae muscles [364] and, in contrast to the positive effect of daily treadmill exercise [365], also exacerbates the shifting of the normal rostral position of CP motoneurons to a more caudal position [364]. Yet, even when CP and TIB nerve branches and not the parent sciatic nerve are surgically repaired to reduce axon misdirection, the normal reciprocal flexor extensor muscle activation is replaced by their coactivation [364,365,366,367].

### 5.7. Conditioning Lesion and Conditioning ES

We reported that ES of an intact adult sciatic nerve performed 7 days prior to excision of DRGs enhanced their neurite outgrowth (Figure 16; [368]), akin to that seen after a conditioning lesion (CL). A CL is a crush of an intact nerve 1–7 days prior to a more proximal nerve crush or transection injury that elevates cAMP levels and increases both the outgrowth of regenerating axons and their regeneration rate [369,370,371,372,373,374,375,376,377]. Thereafter, the question was asked of whether such a conditioning ES of the intact nerve prior to nerve injury and repair may mimic the effects of a CL on nerve regeneration *in vivo* [378,379,380,381,382,383,384,385]. The comparisons of CES and ES revealed that the former, referred to in their publications as functional ES, was superior to the latter [378,379,380,381,382,383,384,385]. The CES increased the numbers and the distance over which neurofilament-positive regenerating axons grew into the distal stump to promote muscle and intraepidermal skin reinnervation [379,380,381,382,383,384,385] in crushed [376,377] and transected and coapted nerves [378,379], after immediate [380] and delayed nerve repairs [382], and after bridge repairs of transected nerve stumps [385]. The CES also led to superior recovery of motor and sensory function [377,378,379,380,381,382,383,384,385]. The authors pointed out that CES could be applied in delayed repair of chronic nerve injuries after major polytrauma that necessitates emergency life or limb management [Section 3], but most require two consecutive surgeries for electrode placement prior to surgical repair. The same issue arises in distal nerve transfer surgery where a ‘donor’ nerve branch of a redundant muscle is cut and sutured to the distal stump of a non-functional ‘recipient’ nerve to restore function [386,387,388,389,390,391]. Oberlin’s transfer to restore elbow flexion is a classic example with a transected ulnar nerve fascicle supplying the flexor carpi ulnaris muscle, sutured to the distal stump of the musculocutaneous nerve branch to biceps brachii muscle [386].

It remains to be determined if CES could be applied to end-to-side neurorrhaphies where an intact nerve fascicle is inserted into a denervated distal stump at the time of the repair of the original nerve repair. The procedure aims to ‘protect’ the stump to enable, particularly over long distances, more successful regeneration and, in turn, muscle reinnervation [392,393,394,395,396]. Nerve regeneration through a denervated rat facial nerve was enhanced by ‘protection’ with end-to-side sensory nerves [395,396,397]. Another CES application may be in side-to side neurorrhaphies [398,399,400,401]. In rats where one or three nerve autografts were placed as cross-bridges between an intact nerve and a chronically denervated distal nerve stump, significantly more motor and sensory neurons regenerated axons after delayed nerve repairs [400].

### 5.8. Drug-Induced cAMP Elevation Mimics the Effect of Electrical Stimulation (ES) on Nerve Regeneration

Administrating rolipram, a specific PDFE4 inhibitor that raises cAMP [402] to a repaired CP nerve, mimics ES in promoting axon regeneration and muscle reinnervation (Figure 17; [403]). These findings are consistent with previous investigations demonstrating that elevating cAMP with forskolin, a stimulant of adenylate cyclase that generates cAMP, increases the rate of peripheral nerve regeneration [311] and that forskolin and dcAMP together promotes spinal nerve regeneration through an SC graft [404].

### 5.9. ES Promotes Sensory Nerve Outgrowth in the Central Nervous System

The high levels of cAMP in neonatal CNS neurons permit them to regenerate their axons after injury in contrast to adult CNS neurons that do not [405,406]. That cAMP elevation by a CL of the sciatic nerve promotes the regeneration of transected sensory nerve fibres in the spinal cord, the elevated cAMP mimicking the CL *in vivo* and *in vitro* [375,407,408], prompted our investigation of the efficacy of ES of the intact sciatic nerve in promoting regeneration of their transected sensory nerves in the CNS. Using CTB to label and visualize axon outgrowth across the T8 lesion site, ES of the sciatic nerve *in vivo* at 20 Hz but not 200 Hz elicited outgrowth but did not accelerate regeneration rate in contrast to a CL that stimulated both outgrowth and regeneration rate (Figure 18; [403]). The high-frequency ES was tested because the large sensory neurons transmit at high frequencies, albeit transiently [409].

Whilst a CL or 20 Hz ES both elevated cAMP equally in DRG neurons and promoted a significant increase in the axon outgrowth across a spinal cord lesion, the CL promoted significantly longer extensions of the axons than the 20 Hz ES [403]). This finding is quite consistent with the known effect of the CL increasing axonal outgrowth and lengthening [369,370,371,372,373] in contrast to the ES effect of promoting only axonal outgrowth [21]. The discrepancy between the findings of the same cAMP elevation by both CL and ES, with the greater effect of the CL in promoting CNS axon outgrowth, indicates that the CL effect is mediated, at least in part, by other mechanisms. This includes non-neural cell contribution. A possible molecular candidate to account for the reliance of the greater efficacy of the CL is oncomodulin, a small calcium-binding protein whose mRNA is highly expressed in neutrophils, the first responders of the innate immune system [410,411] that are recruited rapidly into an injured eye [412]. Oncomodulin binds to retinal ganglion cells with high affinity in a cAMP-dependent manner, increases the level of phosphorylated *CREB*, and stimulates the cells to regenerate long axons beyond the site of the optic nerve injury [413]. The outgrowth was more extensive than other known trophic agents. Moreover, oncomodulin also acts upon peripheral sensory neurons to promote neurite outgrowth, although neutrophils do not participate in the response of the neurons to injury [413].

### 5.10. ES Signaling Pathways

A schematic representation of the intracellular signaling pathways activated by ES is given in Figure 19. ES elevates cAMP in axotomized neurons [403] that, via PKA and CREB [101], initiate expression of neurotrophic factors and their receptors, and of the cytoskeletal proteins, *tubulin* and *actin*, as examples of regeneration-associated proteins. Thereby, ES promotes axon outgrowth to accelerate regenerating nerves towards their denervated targets. Neurotrophins feedback to amplify the cAMP pathway with for example, BDNF-mediated Erk activation producing a transient inhibition of PDE4 activity [414]. Thereby, cAMP is elevated to threshold levels and, in turn, promotes axon outgrowth after immediate or delayed nerve repair of injured peripheral nerves. The positive effect of ES in promoting axonal regeneration after delayed nerve surgery is likely due to the release of mitogens such as neuregulin from the stimulated proximal nerve stump. These mitogens and possibly nutrients available at the site, promote SC proliferation and migration, shifting SCs from their atrophic state to a growth-permissive state. SC neurotrophic factors and p75NTR may present the factors to the regenerating axons to further enhance their growth. Details of the pathways of the efficacy of ES in promoting nerve regeneration and target reinnervation await further experimentation.

### 5.11. ES Accelerates Human Nerve Regeneration and Muscle Reinnervation

Studies of the efficacy of ES in promoting nerve regeneration and muscle reinnervation in rats were extended to patients suffering severe carpal tunnel syndrome (CTS). In these patients, the number of functionally intact MUs in the muscles of the thenar eminence was ~50%. of the normal mean number (+standard error) of 288 + 23 [415]. ES immediately after carpal tunnel release surgery (CTRS), which released the ligament over the crushed median nerve, had a profound positive effect of promoting all the axotomized nerves to regenerate and reinnervate the muscles of the thenar eminence within 6–8 months when the small increase in the reinnervation by the corresponding unstimulated nerves in the control group of patients had not reached statistical significance (*p* > 0.05) even 12 months after CTRS (Figure 20; [416]). This is particularly striking considering that over a distance of 100 mm, the regenerating axons should reach the median eminence musculature within ~3–4 months at a reported rate of 1 mm/day [248,250] for human motor and sensory nerves, respectively. Chan and his colleagues went on to demonstrate the significant acceleration of sensory nerve regeneration in human subjects after surgical repair of transected digital nerves in the hand [417] as well as after ulnar nerve compression at the elbow [418]. In summary, the demonstrated efficacy of low-frequency ES to accelerate nerve regeneration in human subjects as well as in animals, indicates the potential to promote successful outcomes after injuries in patients.

## 6. Conclusions

The rapid decline in electrical activity in motor and sensory nerves during the first month after their injury is associated with and likely caused by the loss of both excitatory and inhibitory synapses on the motoneurons in the spinal cord and by the disconnection of the sensory neurons from their input from the skin and muscles. The concurrent decline in the regenerative capacity of the axotomized neurons and the support of denervated SCs with their declining expression of growth-associated-genes (GAGs) account for the progressive decline in functional recovery with time and distance. Brief ES, by promoting axon outgrowth, accelerates nerve regeneration and target reinnervation after both acute and chronic nerve injuries in animals and in humans. The same ES applied prior to a nerve injury acts as a conditioning ES (CES) to promote both axon outgrowth and the regeneration rate. The ES effect after acute injury is accounted for by (1) preventing the loss of synaptic contacts on axotomized motoneurons, (2) GAG upregulation in response to increased intracellular cAMP, and (3) feedback of expressed neurotrophins that amplify the expression of cAMP. After chronic nerve injuries where synaptic contacts on the motoneurons have already been withdrawn, the ES-induced increase in cAMP, via *CREB* and GAG expression, promotes axonal regeneration and target reinnervation. The mitogens, including neuregulin, that are released from regenerating axons, increase SC proliferation and their transition to a growth-permissive state. Thereby, they amplify the neuronal growth response. 

The ES efficacy has potential for clinical application. CES may be applicable to enhance nerve regeneration in nerve transfer surgeries and end-to-side neurorrhaphies, should open surgery be unnecessary to apply electrodes for CES. Finally, our findings that delayed nerve repair is deleterious for functional recovery emphasize the importance of ensuring minimal delay between the diagnosis of nerve injury, repair, and ES application. Thereby, we might anticipate improved outcomes of nerve surgeries in the future.

## Figures and Tables

**Figure 1 ijms-25-00665-f001:**
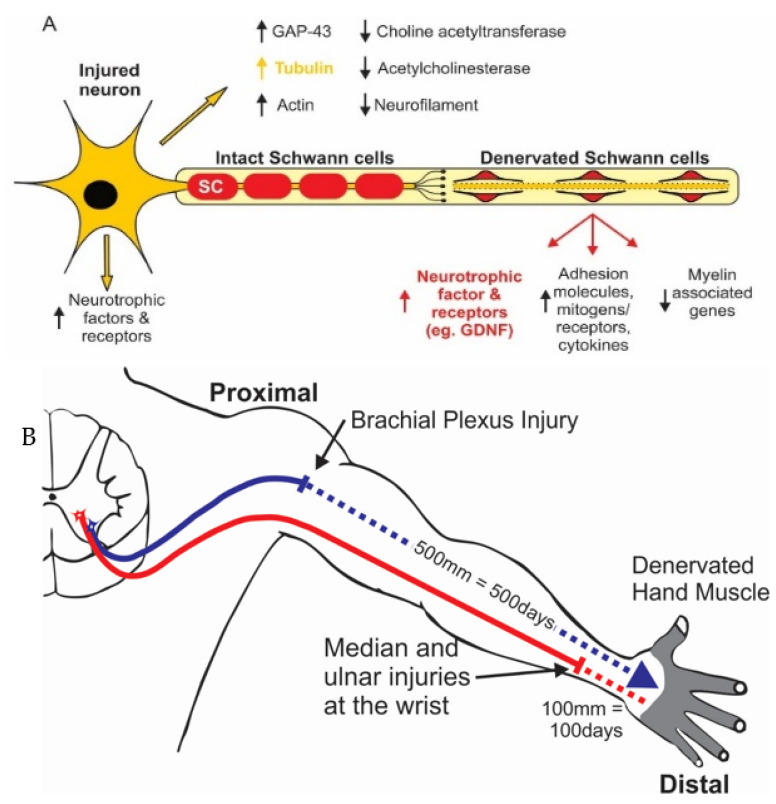
Diagrammatic illustrations of (**A**). the downregulation of transmitter-associated enzymes and the upregulation of regeneration-associated genes in axotomized motoneurons and the downregulation of myelin-associated genes and other molecules of the intact nerve after denervation of the distal stump. Denervated Schwann cells elongate and support the regeneration of axons after they sprout from the proximal stump of the transected nerve. (**B**). Diagrammatic illustration of axonal regeneration over long distances in human limbs. Axons that regenerate at 1 mm/day will not reach denervated target muscles and sense organs for at least 500 days after a brachial nerve injury (downward arrow). Even median and ulnar nerve injuries at the level of the wrist (upward arrow) will require months for regenerating axons to reach the denervated targets. The regeneration rate was determined with the Tinel sign that identifies the site at which a tap on the regenerating nerve elicits a tingling sensation in a conscious patient. Modified from [60].

**Figure 2 ijms-25-00665-f002:**
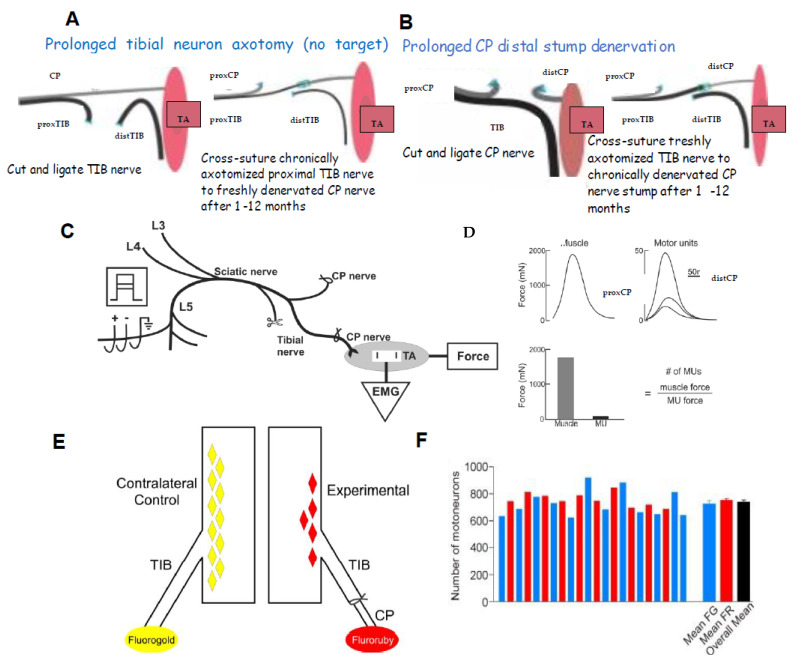
Physiological and morphological measures of regenerative success after nerve injuries. Diagrammatic illustration of surgeries in *Sprague-Dawley* rats for (**A**). prolonging proximal tibial motoneuron (proxTIB) axotomy, leaving the distal tibial nerve denervated (distTIB), and the common peroneal nerve (CP) intact; and (**B**). prolonging CP distal nerve stump (distCP) and tibialis anterior muscle denervation and leaving the TIB nerve intact. Each of the nerve stumps were sutured to innervated muscle to prevent nerve regeneration for up to 12 months prior to refreshing the chronically axotomized proxTIB nerve stump to cross-suture it to the freshly denervated distCP. (**C**). Recordings were made at least 5 months later of the electromyographic (EMG) and the isometric contractile forces of the innervated tibialis anterior (TA) muscle and single MUs (motor units) in response to 2× threshold stimulation of the regenerated TIB nerve and the isolated ventral root filaments. Examples of muscle and MU twitch contractions are shown, and their contractile forces plotted in the histogram. The number of MUs, and hence, the number of motoneurons that reinnervated the TA muscle was calculated as the ratio of the muscle and MU forces. Retrograde labelling of (**D**) TIB motoneurons that regenerated their nerve fibres into the CP distal nerve stump and (**E**). TIB motoneurons retrogradely labelled from the TIB nerve (**F**). The numbers (+standard error) of the motoneurons retrogradely labelled with FG or FR in several different control experiments designed to determine whether FG and FR dyes were as effective as each other in counting motoneurons projecting their nerve fibres to the point of dye application. Mean numbers were not significantly different (*p* > 0.05) such that either dye was used to count the motoneurons as well as sensory neurons that regenerated their axons after experimental nerve injuries. Adapted from [60].

**Figure 3 ijms-25-00665-f003:**
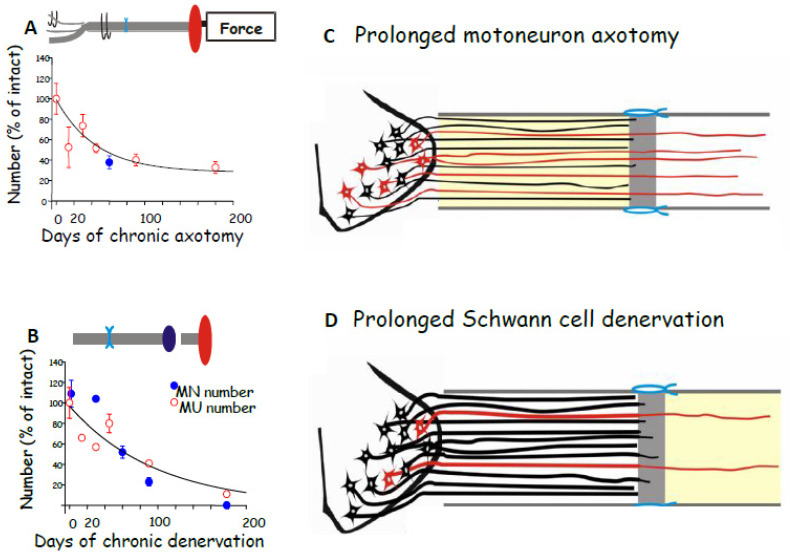
(**A**,**B**). The exponential decline in the regenerative capacity of rat lumbosacral tibial (TIB) motoneurons (MNs) after chronic axotomy and prolonged Schwann cell (SC) denervation in the common peroneal (CP) distal nerve stump. The suture of the proximal TIB nerve stump to the distal CP nerve stump (cross-suture) was delayed for up to 6 months after cutting the TIB nerve prior to resuture to a freshly cut CP nerve (chronic axotomy) or after cutting and delaying the suture of the cut distal CP nerve to the freshly cut TIB proximal nerve stump. The Y-axis, the regenerative success (%), plots the mean (+standard error) number of reinnervated MUs in the tibialis anterior muscle (motor units (MUs), open red circles), and of TIB motoneurons that regenerated their axons into the CP distal nerve stump (closed blue circles), expressed as a percentage of the numbers in unoperated rat hindlimbs, and plotted as a function of the durations of A. chronic TIB axotomy and (**B**). chronic CP denervation. The effects are shown figuratively in (**C**,**D**), respectively, for regeneration 4 months after TIB–CP nerve cross suture. The motoneurons and their axons are shown in black and red for those that did NOT and those that DID regenerate axons into the distal nerve stump, respectively. The methods used to determine regenerative success are shown in Figure 2. Adapted from [60].

**Figure 4 ijms-25-00665-f004:**
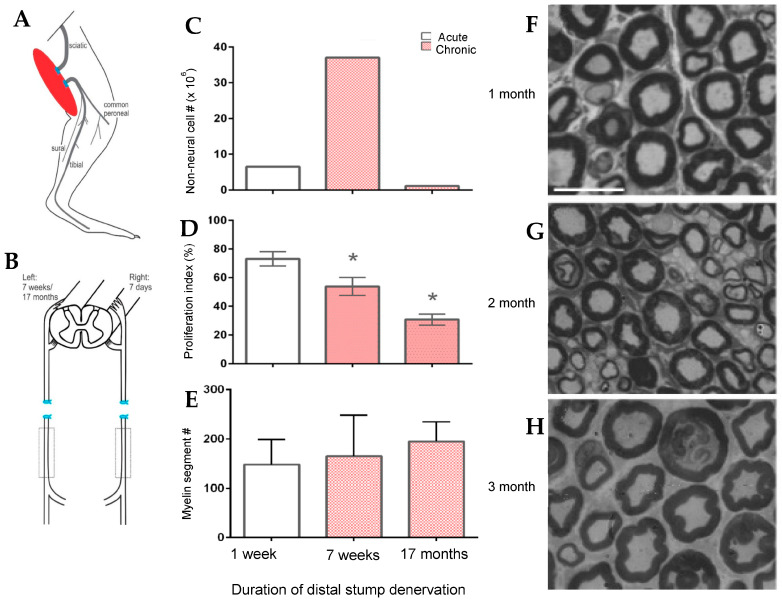
Survival of chronically denervated Schwann cells (SCs) and their capacity to proliferate, myelinate neurites and nerve fibres *in vitro* and *in vivo*, respectively, and to support recovery of reinnervated nerve fibre size. (**A**). Sterile surgery to prolong SC denervation in the rat sciatic distal nerve stump by cutting the nerve and ligating the proximal and distal nerve stumps to innervated muscles. (**B**). The left sciatic distal nerve stump remained denervated for either 7 days, 7 weeks, or 17 months and the right sciatic nerve was denervated for 7 days for direct comparison of the data of acutely denervated nerves with the data from chronically denervated nerves. Histograms of the mean (+standard error) of (**C**). numbers of non-neural cells from the sciatic distal nerve stump after chronic denervation that increased significantly after 7 weeks and declined to low levels by 17 months (* *p* < 0.05, # refers to the number of non-neural cells), (**D**). numbers of ^3^H-thymidine positive Schwann cells undergoing proliferation in co-cultures of dorsal root ganglion (DRG) sensory neurons that declined progressively with the duration of chronic denervation, and (**E**). the myelinated segments extended by co-cultures of the non-neural cells and DRG neurons.(# refers to the number of non-neural cells) (**F**). Full recovery of tibial nerve size and myelin thickness after regeneration 1–3 month after chronically denervated common peroneal distal nerve stumps *in vivo*. For (**F**–**H**), the scale bar = 25 µm. Adapted from [267].

**Figure 5 ijms-25-00665-f005:**
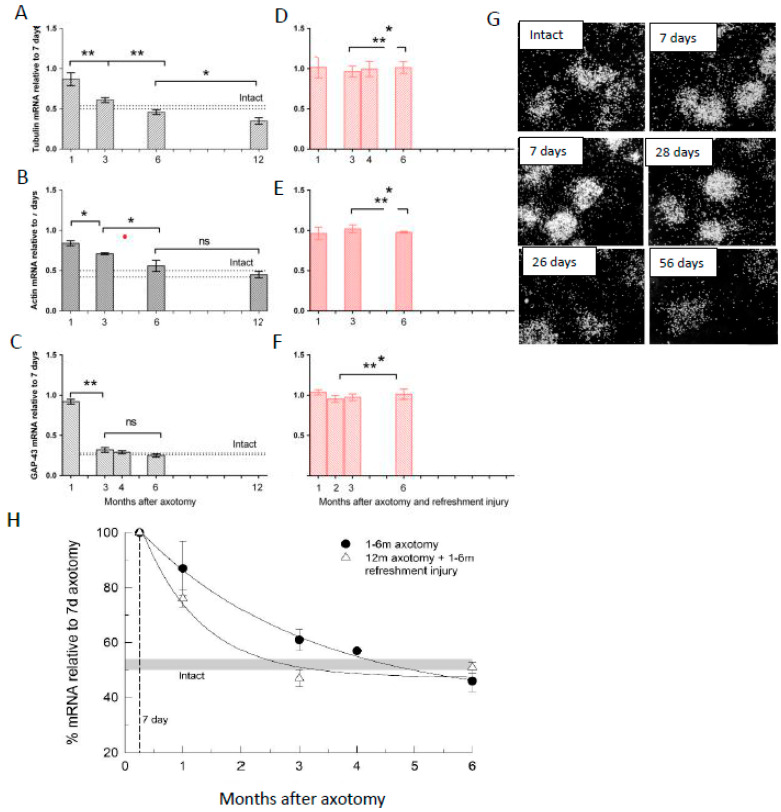
Expression of mRNA of cytoskeletal proteins in sciatic motoneurons decays with time after chronic axotomy. Time-dependent decay in the *in situ* hybridization signals of (**A**)*. tubulin,* (**B**). *actin*, and (**C**)*. GAP-43* mRNA (on the Y-axis) as a function of months after axotomy. A refreshment injury irrespective of the duration of axotomy elevates the expression of (**D**). *tubulin*, (**E**). *actin*, and (**F**). *GAP-43* to maximum levels. The mRNA levels in the chronically axotomized motoneurons were expressed relative to the high levels of mRNAs that were detected at a week after axotomy of the contralateral sciatic nerve (**A**–**G**). The high levels of the *in-situ* hybridization signals detected in sciatic motoneurons 7 days after sciatic nerve transection and ligation decay with time. (**H**). The elevated *tubulin* mRNA levels after a refreshment injury decayed more rapidly than after a single nerve transection. The scale bar at the bottom right image is 20 µm. Mean values + standard error bars are shown with significance at * *p* < 0.05 and ** *p* < 0.01 denoted by 1 and 2 *s. Where the values were not significant with *p* < 0.05, not significant is denoted by the abbreviation of ns. Adapted from [291].

**Figure 6 ijms-25-00665-f006:**
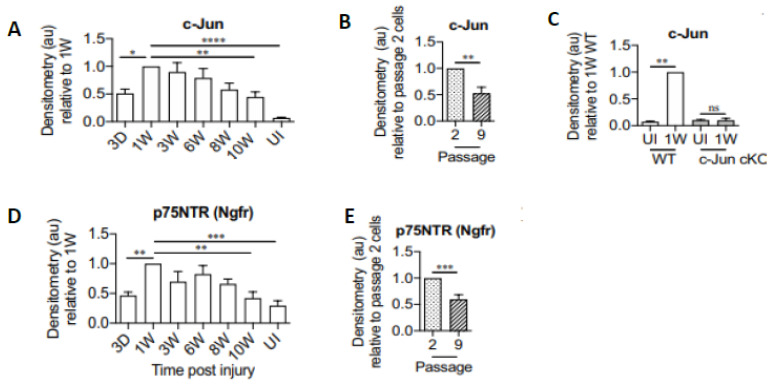
Protein levels of the transcription factor *c-Jun* and the neurotrophic factor receptor (p75NTR) in mouse Schwann cells (SC) decline with time after sciatic nerve chronic denervation. The quantification of representative Western blots with densitometry of *c-Jun* protein in (**A**). uninjured (UI) nerve and distal nerve stumps 1–10 weeks after nerve ligation, normalized to the data at 1 week post-injury, (**B**). SC cultures after two and nine passages, and (**C**). 1 week after injury in wild-type (WT) and *c-Jun* conditional knockout (cKO) mice, and of p75NTR, (**D**). with time after injury, and (**E**). in SC cultures after two and nine passages. The *c-Jun* located to the denervated SCs was verified in *c-Jun* cKO mice where SC *c-Jun* was inactivated selectively. Mean values + one standard error bar are shown with significance at * *p* < 0.05, ** *p* < 0.005, *** *p* < 0.001, and **** *p* < 0.0001 denoted by 1, 2, 3, and 4 *s. Adapted from [293].

**Figure 7 ijms-25-00665-f007:**
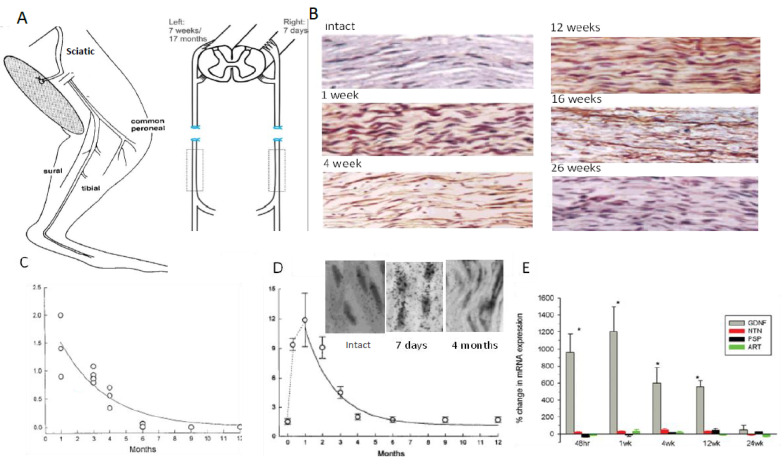
The exponential decline in neurotrophic factor receptor *(p75NTR*) mRNA and protein in rat Schwann cells (SCs) and the expression of glial cell neurotrophic factor *(GDNF*) in denervated distal nerve stumps decline as a function of time after chronic denervation. (**A**). Rat sciatic nerve section and ligation of the proximal and distal nerve stumps to nearby innervated muscle to prevent nerve regeneration. (**B**,**C**). Progressive decline in the relative density of p75NTR immunohistochemistry with time of chronic SC denervation. (**D**). Decline in the mean (+standard error) of the *p75NTR* mRNA, depicted as dark grains following in situ hybridization histochemistry in intact sciatic nerve and transected distal nerve stumps (see insert), with the duration of chronic denervation (months). (**E**). The mRNA levels of *GDNF*, Neurturin (*NTN*), Persiphin (PSP), and Artemin (*ART*) mRNAs measured by semiquantitative PCR with *GAPDH* as an internal control and expressed as the change in the transected distal nerve stumps vs the sham-operated control sciatic nerves, plotted as a function of chronic denervation. * *p* < 0.05 Adapted from [224,290].

**Figure 8 ijms-25-00665-f008:**
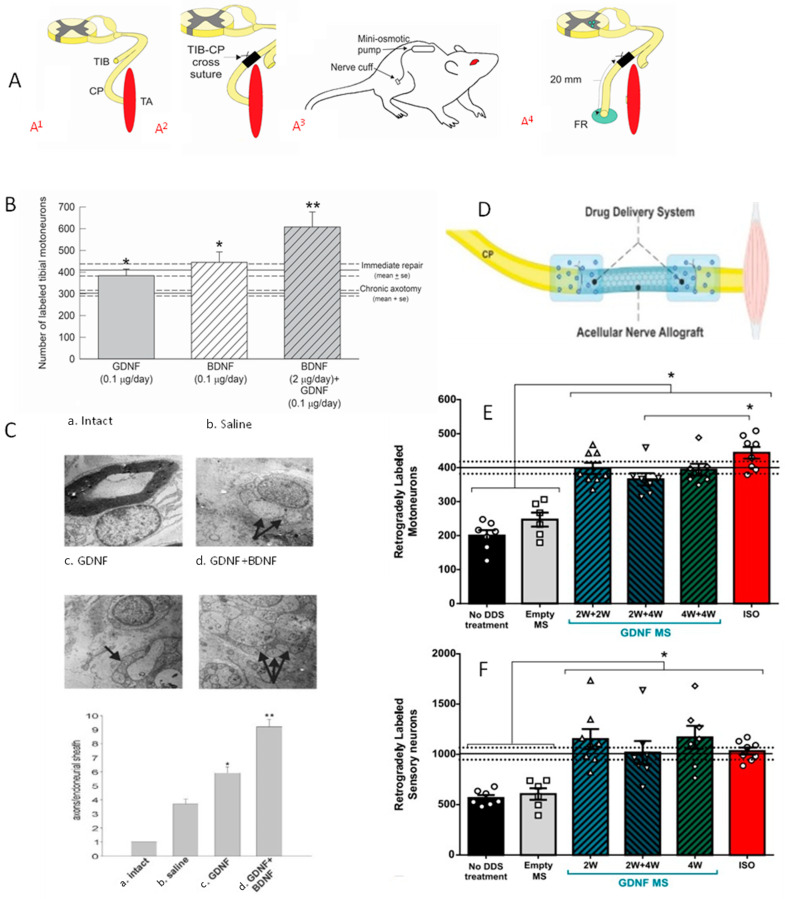
Exogenous delivery of brain- and glial-cell-derived neurotrophic factors, BDNF and GDNF, respectively, was effective in improving nerve regeneration after chronic axotomy, but it increased axon sprouting in the distal nerve stump. (**A**,**A^1^**). The TIB (tibial nerve) in the rat was cut and ligated. (**A^2^**). After two months, the proximal stump of the TIB nerve was cross-sutured to the freshly denervated CP (common peroneal) nerve stump, and (**A^3^**). a mini-osmotic pump was placed on the back of the rat for constant infusion of low dose BDNF (2 µg/day for 28 days). (**A^4^**). Following a 2-month period of chronic axotomy, fluororuby (FR) was administered to the CP distal nerve stump 20 mm from the cross-suture site for retrograde labelling of the TIB motoneurons that had regenerated their axons into the denervated CP distal nerve stump. (**B**). The mean (+one standard error (SE)) of the number of motoneurons that regenerated their axons counted in all 50 µm longitudinal sections of the ventral horn of the spinal cord and with a correction factor of 0.6 applied, were significantly reduced two months after chronic axotomy. Both GDNF and BDNF were effective in increasing motoneuron regeneration to normal levels and the combination of factors elevated the levels even more. (**C**). Transmitting electron micrographs of Schwann cells (SCs) and their axons revealed (**a**). one SC per intact nerve, but (**b**–**d**). increased numbers of SCs around each regenerated nerve in the distal nerve stump 2 months after TIB–CP cross-suture and administration of (**b**). saline, (**c**). GDNF, or (**d**). GDNF and BDNF administration via a cuff around the suture site linked to a mini-osmotic pump. The increased numbers of SCs around each regenerated nerve are pointed out by arrows in the micrographs. (**D**). Local delivery of GDNF to the suture sites of an acellular nerve allograft (ANA) interposed between the CP proximal and distal nerve stumps. GDNF was encapsulated with an efficiency of 78 + 3% and GDNF loading of 0.72 + 0.08 µg/mg of microspheres (MS; diameters of 45 + 5 µm), placed onto the suture sites, and held in place by surrounding them with 2 µL gels placed below and above the suture sites that adhered to one another. (**E**). The mean (+SE) number of retrogradely labeled CP motoneurons (retrogradely labelled 10 mm from the ASA eight weeks after surgery) that regenerated their axons through the ANA plotted for control conditions of no delivery system (DDS) and empty MSs and experimental conditions of GDNF delivery in combinations of 2- and 4-week formulations. (**F**). Numbers of sensory neurons counted in every fifth dorsal root ganglia 20 µm section. Irrespective of the nature of GDNF delivery, all motor and sensory neurons regenerated their axons through the ANA in eight weeks as compared to ~50% of them that regenerated through the ANAs that did not deliver GDNF. Statistical significance at *p* < 0.05 and *p* < 0.01 is denoted by a * and a **, respectively. Adapted from [256,318].

**Figure 9 ijms-25-00665-f009:**
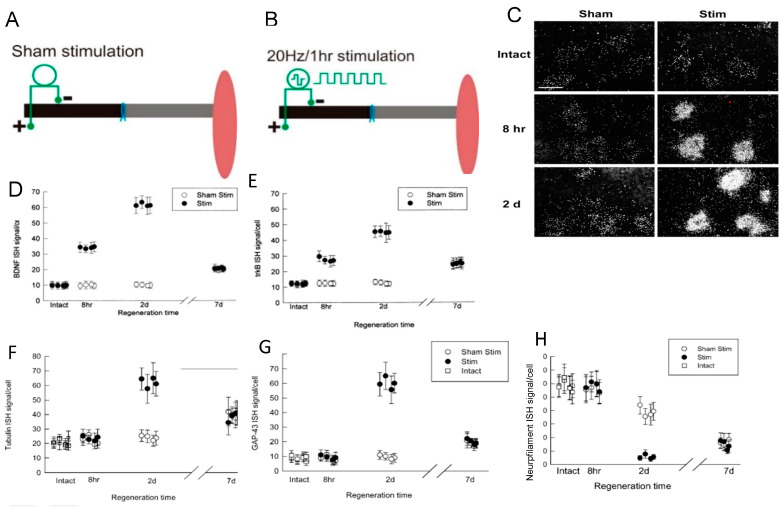
The time course of gene expression of *BDNF* (brain-derived neurotrophic factor) and its *trkB* receptor, and of the regeneration-associated genes, *tubulin*, and *GAP-43*, after axotomy. Transection and surgical repair of the rat femoral nerve and (**A**). sham or (**B**). 1 h 20 Hz electrical stimulation (stim) of the proximal stump. (**C**). Dark-field micrographs of in situ hybridization (ISH) with 35S-labeled oligonucleotides complementary to *tubulin* after sham and stim. Scale bar, 50 µm. Means + standard deviation of ISH signals/motoneuron of (**D**)*. BDNF*, (**E**). *trkB*, (**F**). tubulin, (**G**). *GAP-43*, and (**H**). *neurofilament* in individual rats are plotted as a function of time after femoral nerve axotomy after sham and stim. Adapted from [75].

**Figure 10 ijms-25-00665-f010:**
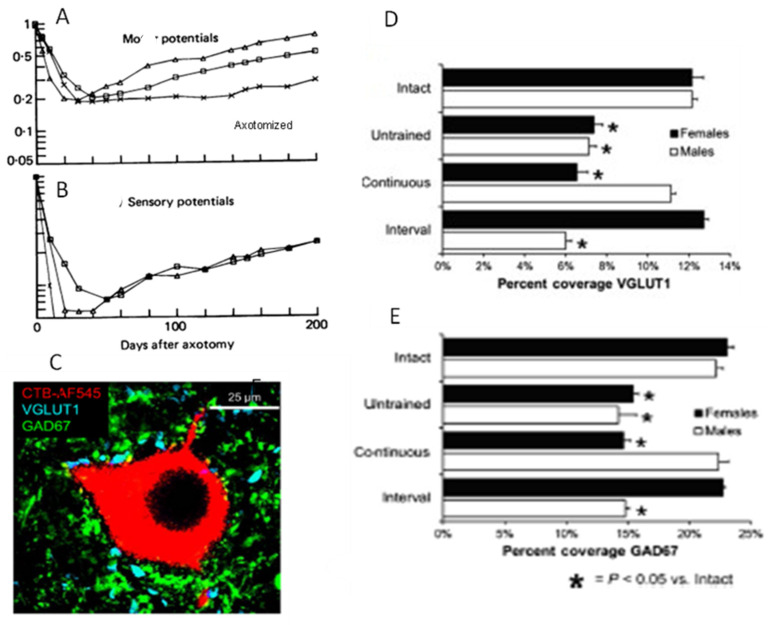
(**A**) decline in electrical activity in motor nerves after axotomy coincides with a loss of synaptic contacts from the axotomized motoneurons and increased motor activity restores the synaptic contacts. The relative amplitude of (**A**). motor and (**B**). sensory cross-correlation peaks normalize to pre-operative values, decline with an exponential time course over the first month of axotomy and begin to recover when nerve regeneration is encouraged after nerve repair by resuture of cut nerve stumps (open triangle) and by suture of proximal nerve stumps to denervated muscle (open boxes), but reach a stable plateau if nerve regeneration is prevented by nerve ligation (x’s). (**C**). A one µm thick confocal image of a retrogradely CTB-AF545-labeled motoneuron with immuno-histochemical visualization of VGLUT1 (vesicular glutamate transporter 1)-containing excitatory synapses, arising mainly from primary afferent neurons, and GAD67 (glutamic acid decarboxylase 67)-containing inhibitory synapses were used to measure the percentages of the neuronal perimeter in contact with the synapses. The silencing of motoneurons by axotomy is due to the withdrawal of (**D**). glutamatergic and (**E**). GABAergic synaptic contacts from the neurons. The withdrawal of both synapses is prevented by continuous treadmill exercise in males and by interval training in females. Adapted from [58,319].

**Figure 11 ijms-25-00665-f011:**
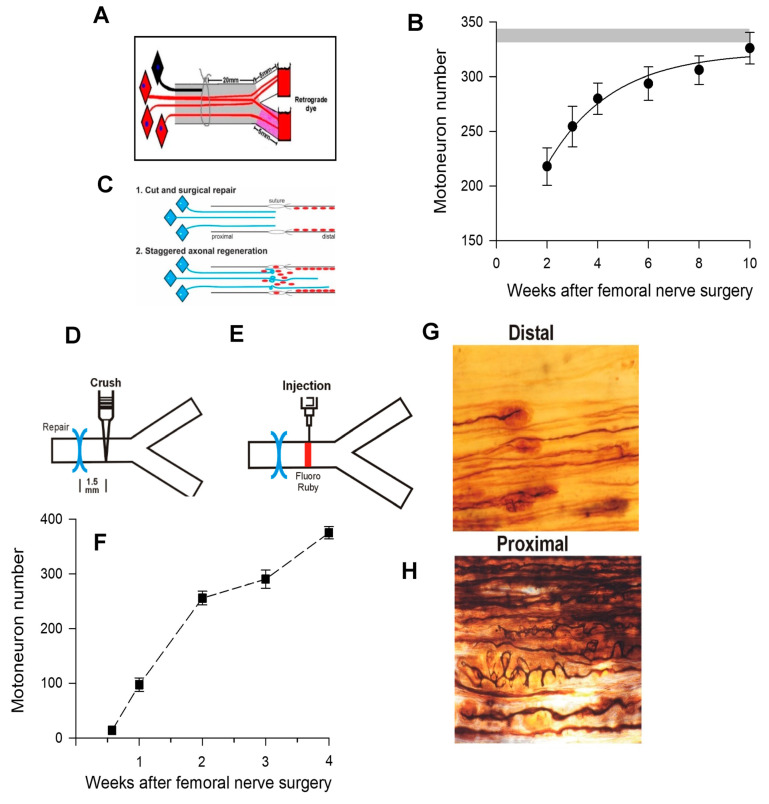
Motor nerve regeneration is staggered. (**A**). The femoral nerve in the rat was cut 20 mm from its branches to the quadriceps muscle and to the skin. It was repaired immediately with a microsurgical technique. The regeneration of axons into the two branches was evaluated 2 to 10 weeks later by application of fluorescent dyes, RR (rubyred) for 1–2 h to one branch and FG (fluorogold) to the other branch, 5 mm from the point of their division from the femoral nerve. (**B**). Motoneurons that were retrogradely labelled with either or both dyes are the sum of those that regenerated their axons into either branch as well as both the branches. The number of these motoneurons that regenerated their axons over the distance of 25 mm was ~55% of all the motoneurons in the femoral motoneuron pool, the number increasing with an exponential time course to reach the 350 motoneurons of the control intact nerve (shown by the shaded horizontal lines that represent + standard errors of the mean values). The 10 weeks required for all the motoneurons to regenerate the 25 mm distance to the site of dye application was surprisingly long when a latency of at least 2 days and a regeneration rate of 3 mm/day is considered [334]. Based on the latency and regeneration rate, a 2-to-3-week period of regeneration would be expected for all the motoneurons to regenerate their axons. (**C**). The possible explanation that axon outgrowth across the surgical site is staggered, with some axons even growing backwards into the proximal nerve stump, proved to be correct. The Schwann cells that multiply after nerve injury and line the distal nerve stump are known to migrate into the injury site and are shown in red. (**D**). To determine whether this explanation accounts for the observed slow regeneration of nerve fibres, those axons that had regenerated just across the suture line of the femoral nerve repair site were retrogradely labelled by crushing the distal nerve stump 1.5 mm from the surgical site for (**E**). microinjection of FR. (**F**). The number of the retrograde labelled motoneurons that regenerated the axons increased slowly over a period of 4 weeks after femoral nerve repair surgery. Silver staining of (**G**). staggered the regeneration of axons in the distal stump, and (**H**). the tortuous paths taken by the axons within the proximal stump are consistent with Cajal’s early findings and with later observations of the outgrowth of fluorescent axons from the proximal stump of thy-1-YFP transgenic mice into a denervated distal nerve stump of a non-transgenic wild-type litter mate [28,58]. Adapted from [58].

**Figure 12 ijms-25-00665-f012:**
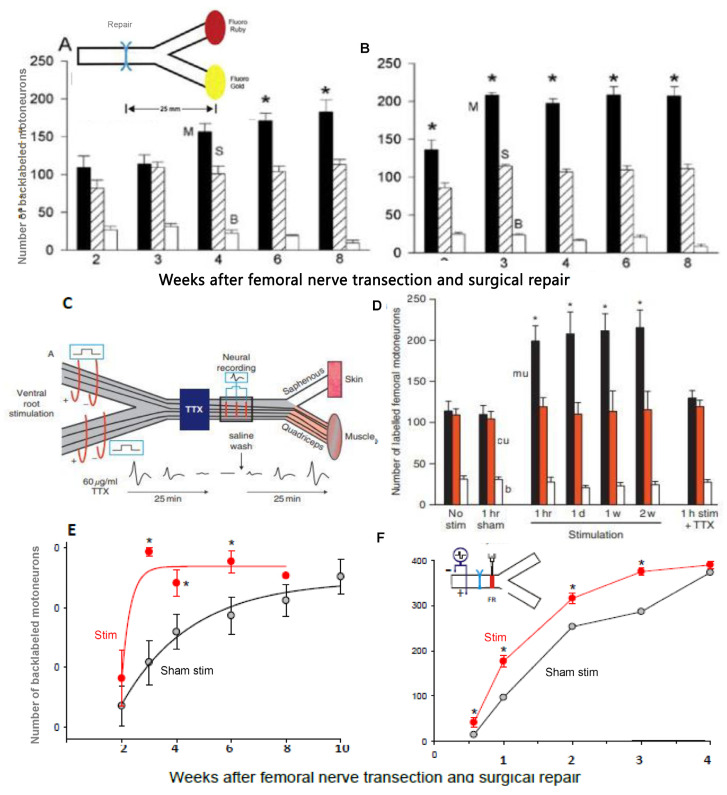
Low-frequency (20 Hz) electrical stimulation (ES) promotes nerve regeneration by accelerating axon outgrowth across the site of transection and surgical repair. (**A**). The number of retrogradely labelled femoral motoneurons (Motoneuron) with fluorogold (FG) and fluororuby (FR: analogous to ruby red) fluorescent dyes applied to the motor quadriceps (M) and sensory saphenous (S) nerve branches, respectively, were counted 2 to 10 weeks after microsurgical repair of the femoral nerve. Following early random reinnervation of both branches and a small number of regenerating axons into both branches (**B**), there was preferential reinnervation of the M branch with a significant (*p* < 0.05) increase in the mean (+standard error) numbers of motoneurons regenerating their axons preferentially into the M branch, occurring over a period of 8 to 10 weeks. The inappropriate regeneration of axons into the S branch remained constant throughout. The delivery of continuous daily 20 Hz ES (100 µs, 2× supramaximal electric pulses, for 1 h, 1, 7, or 14 days) proximal to the repair site, resulted in (**B**). a dramatic and statistically significant (*p* < 0.05) elevation in the number of motoneurons regenerating their axons into the M nerve branch within 3 weeks. The efficacy of the ES depended on the retrograde conduction of action potentials to the motor and sensory neurons because (**C**). tetrodotoxin (TTX) application proximal to the site of ES that completely blocked the progression of ES-induced action potentials resulted in (**D**). elimination of the ES acceleration of preferential reinnervation of the M branch. (**E**). The dramatic accelerated reinnervation of the two femoral nerve branches is illustrated by the comparison of the total mean number (+SE) of the motoneurons regenerating their axons 25 mm into the M and S branches and both after ES (closed circles) and sham ES (open circles). (**F**). The effect of the ES was to accelerate axon outgrowth across the repair site because the mean number (+SE) of femoral motoneurons that regenerated their axons just across the repair site, recorded by injecting FR into the femoral nerve 1.5 mm distal to the repair site (see insert), increased significantly by ES (closed circles) as compared to sham ES (open circles statistical significance at *p* < 0.05 is indicated by * s. Adapted from [20,21,22].

**Figure 13 ijms-25-00665-f013:**
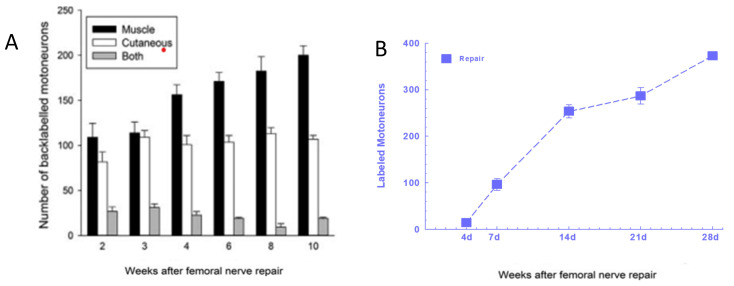
Preferential motor reinnervation of the motor branch of the femoral nerve is accounted for by differential expression of neurotrophic factors in the denervated distal nerve stump. Numbers of rat femoral motoneurons that regenerate their axons (**A**). 25 mm from the site of transection (injury) and surgical into appropriate muscle (black bars), inappropriate sensory cutaneous branches (white bars), and both branches (grey bars), and (**B**). across the repair site. (**C**). The fold-increase in the mRNA of *PTN* (pleiotrophin) and *GDNF* (glial cell-derived neurotrophic factor) in the distal nerve stumps of transected and repaired femoral nerve expressed relative to that of intact nerves increases and decreases as a function of days of the denervation of the distal nerve stump. (**D**). Diagrammatic representation of motoneurons regenerating their nerve fibres into appropriate and inappropriate muscle and cutaneous pathways, respectively, as well as into both. Adapted from [336].

**Figure 14 ijms-25-00665-f014:**
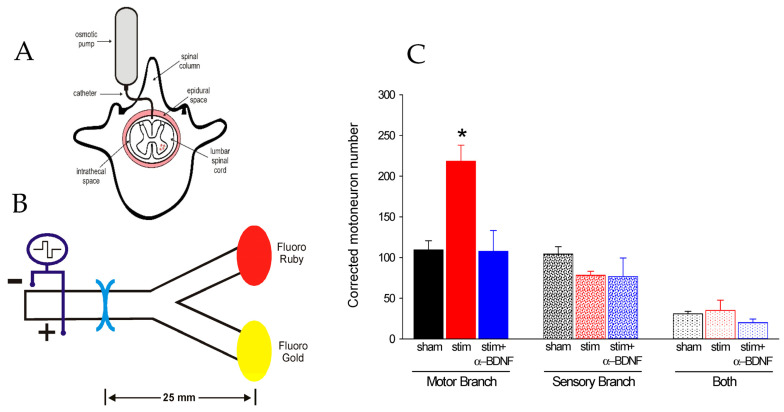
BDNF (brain-derived neurotrophic factor) is essential for the efficacy of ES (20 Hz electrical stimulation for one hour) in accelerating nerve regeneration. (**A**). The infusion of the BDNF antibody, α-BDNF, into the intrathecal space of a rat lumbar spinal cord 3 days prior to (**B**). ES via electrodes placed proximal to the surgical site of femoral nerve transection and microsurgical repair site and the application of the fluorescent dyes, fluororuby and fluorogold, to each of the femoral motor and cutaneous sensory branches for retrograde labelling of those motoneurons that regenerated their axons into either or both the branches at 3 weeks after surgical repair. (**C**). The motoneurons were counted and a correction factor applied. The mean numbers (+standard errors) of motoneurons that regenerated their axons 25 mm into the appropriate and inappropriate motor and sensory nerve branches, respectively, were plotted for the motoneurons subjected to ES (stim) and sham ES. ES accelerated the regeneration of motor nerves into the appropriate motor branch without affecting either the regeneration into the inappropriate motor branch or into both branches (see also Figure 12D,E). This accelerated motor nerve regeneration was blocked by infusion of the α-BDNF. The number of motoneurons that regenerated their axons inappropriately into the sensory branch of the femoral nerve or those that regenerated their axons into both branches was not changed by the infusion of α-BDNF. Control data after α-BDNF administration to sham-stimulated femoral nerves, indicated, but could not prove with our insufficient numbers for statistical analysis, that the antibody infusion had no effect on nerve regeneration. These unpublished data were obtained in collaboration with Drs. Verge and Pettersson at the University of Saskatchewan. Significance at *p* < 0.05 is denoted by a *.

**Figure 15 ijms-25-00665-f015:**
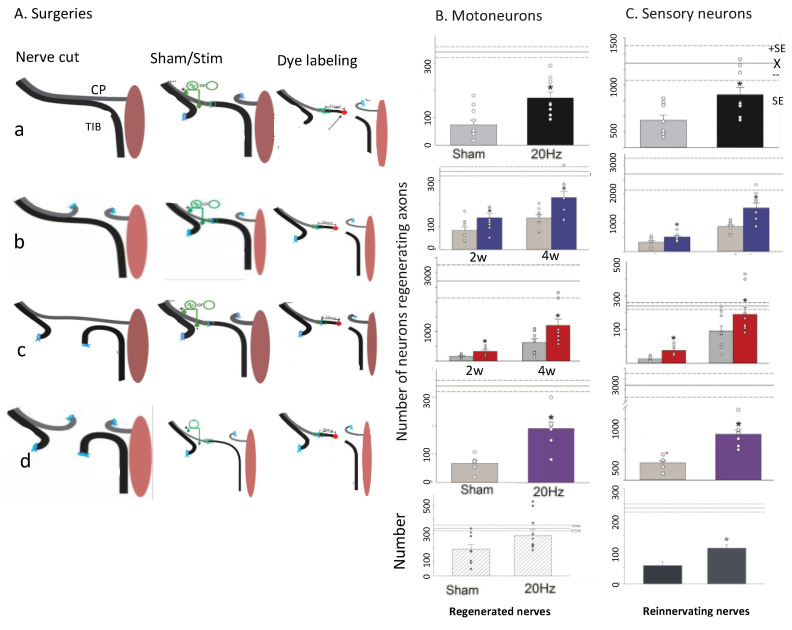
One hour 20 Hz electrical stimulation (ES) promotes nerve regeneration and target reinnervation after delayed nerve repair. (**A**). Surgeries: The common peroneal nerve (CP), and/or tibial (TIB) nerve were cut and cross-sutured either (**a**). immediately or 2 months after (**b**). prolonged CP axotomy and/or distal TIB nerve denervation (**c**,**d**). and the proximal nerve stump was subjected to either sham or ES. Histograms show that the mean (+standard errors) numbers of motoneurons in (**B**,**a**–**d**). and sensory neurons in (**C**). that regenerated their axons 20 mm into the distal nerve stump, 2 and 4 weeks after 20 Hz ES, were significantly greater than the mean numbers after sham ES, the significance of *p* < 0.05 denoted by a *. The different colors used in the histograms in (**a**–**d**) denote data obtained from each of the 4 different surgeries carried out. The open symbols in the histograms show the data collected from each experiment. The lowest and 5th set of histograms in e. show the number of regenerating motor nerves and those that had reinnervated tibialis anterior muscle, (obtained in final acute recording made of muscle and motor units), 5 months after delayed nerve repair. The labels underneath each of the 5 rows of histograms under the heading of (**B**). Motoneurons, apply for the 5 rows of histograms under the heading of (**C**). Sensory neurons. Adapted from [24].

**Figure 16 ijms-25-00665-f016:**
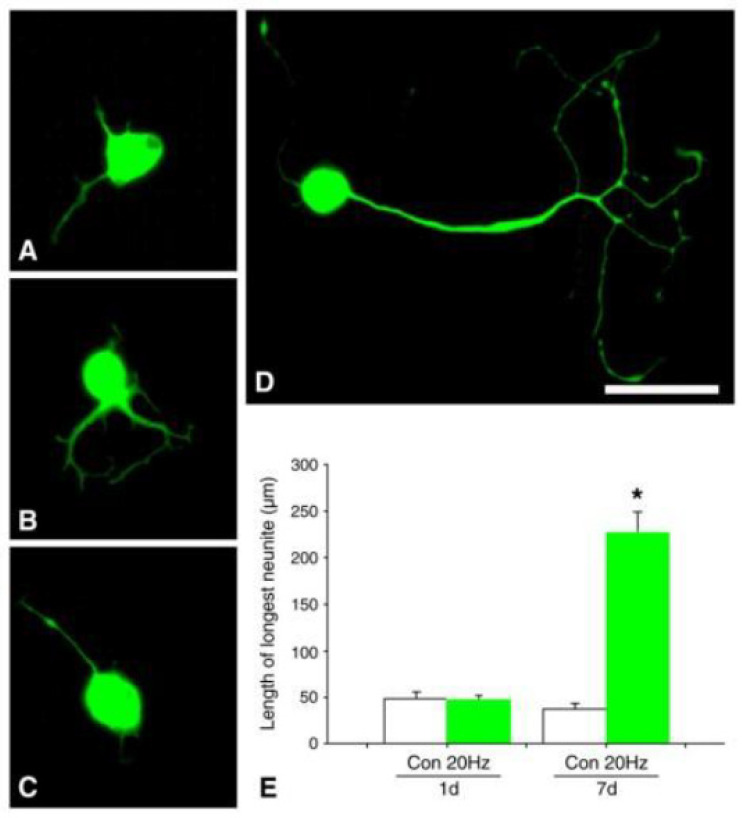
Electrical stimulation of the intact adult sciatic nerve of the rat at 20 Hz for 1 h (ES) *in vivo* before excising dorsal root ganglion (DRG) sensory neurons and plating them on a laminin substrate *in vitro* promotes their neurite outgrowth. An intact sciatic nerve 1 day after (**A**). no ES and (**B**). ES. An intact sciatic nerve 7 days after (**C**). no ES and (**D**). ES. (**E**). The mean + standard error of the lengths of the longest neurites (in µm) for the control and experimental DRG neurons that were not and were stimulated, respectively. The ES of intact sciatic nerves promoted neurite extension from plated DRG neurons. This promoted neurite extension is akin to the effect of a nerve crush conditioning lesion (CL) that is made 7 days prior to the transection and suture of the nerve proximal to the CL, which accelerates nerve regeneration. The neurons were immunostained with β-tubulin III. The calibration bar = 50 µm. Statistical significance at *p* < 0.05 is denoted by a *. Adapted from [368].

**Figure 17 ijms-25-00665-f017:**
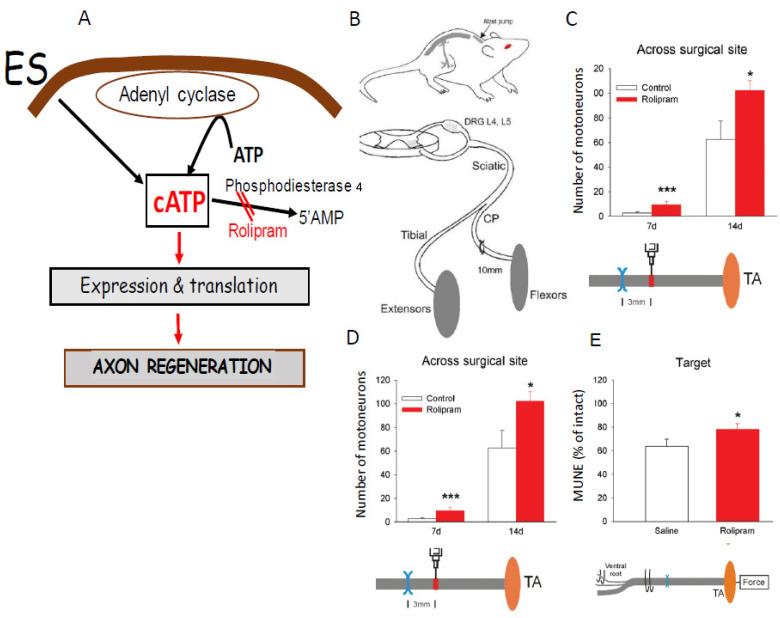
(**A**). We asked the question as to whether cAMP mediates the effect of ES on axon regeneration by administering rolipram to the site of injury and surgical repair to block phosphodiesterase 4 that hydrolyses cAMP. (**B**). Rolipram (1:1 saline/dimethyl sulfoxide [DSMO]) or vehicle (1:1 saline/DMSO) was delivered via an Alzet pump at a rate of 0.4 µmol/g/h to the site of the cut and sutured the CP (common peroneal) nerve that was surrounded by a 3 mm long silicone silastic cuff of 0.7 mm interior diameter. The mean (+standard error) number of CP motoneurons that regenerated across the suture site (expressed as a percentage of the numbers of intact CP motoneurons on the contralateral side) were retrogradely labelled with rubyred (RR) (**C**). 3 mm, and (**D**). 10 mm from the suture site. (**E**). MUNE (the estimated number of MUs (motor units) that equals the ratio of the muscle and mean MU twitch forces of the reinnervated TA (tibialis anterior) muscle after the CP nerve suture repair) expressed as a percentage of MUNE in muscles with intact innervation. There was a significant increase in the number of CP motoneurons that regenerated their axons into the distal nerve stump (**C**). 3 mm past the CP nerve suture site after 7 (*** *p* < 0.01) and after 14 days (*** *p* < 0.01), (**D**). 10 mm past the suture site after 14 days and (**E**). reinnervated TA muscle after 5 months (* *p* < 0.05). Adapted from [403].

**Figure 18 ijms-25-00665-f018:**
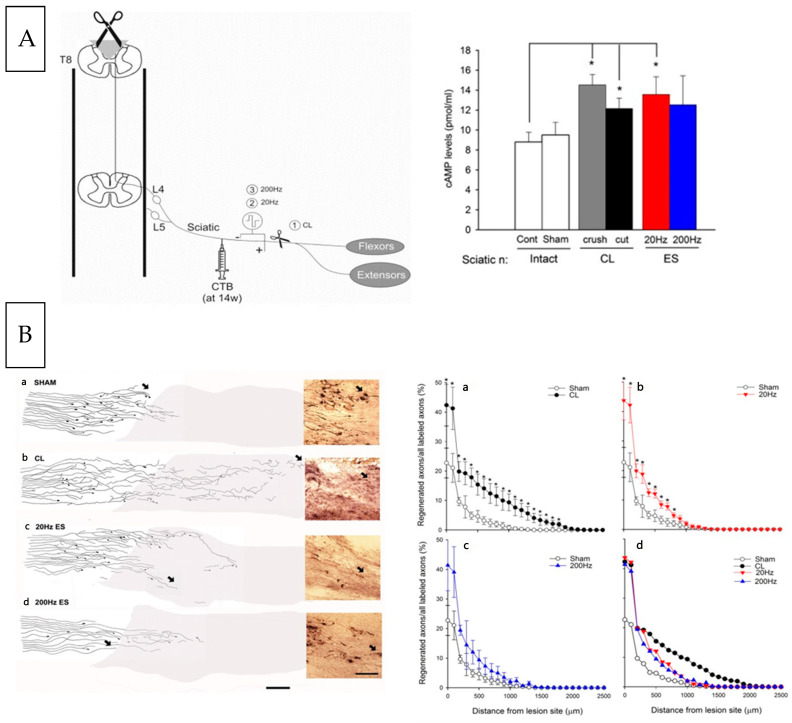
Electrical stimulation (ES) promotes regeneration of transected sensory nerves in the central nervous system (CNS). (**A**). Figurative drawing of the surgical transection at the thoracic eight dorsal column of the central axons of the lumbosacral dorsal root neurons whose peripheral axons in the sciatic nerve were (1) either crushed or cut as the conditioning lesion (CL) or left intact. The intact nerve was electrically stimulated for 1 h at (2), 20 Hz, and (3) at 200 Hz with the ES in-tensity adjusted to 2× the motor threshold for eliciting gastrocnemius muscle contractions. A 1% solution of cholera Toxin B (CTB) was injected into the sciatic nerve 14 weeks later to label CNS sensory nerves *in vivo*. (**A**). Insert. The mean (±standard error (SE)) cAMP levels in L4 to six dorsal root ganglion neurons were significantly raised 24 h after crush and cut CLs and 20 Hz ES but not after 200 Hz, despite the transient high firing rates of sensory nerves. (**B**). The camara lucida drawings of spinal cord 25 µm sagittal sections of CTB-immunocytochemical identification of axons show the longest axons (shown by arrows) growing beyond the lesion site (in grey) after the CL with shorter axons growing after 20 Hz ES. The letters a–d that denote (**a**). sham, (**b**). CL, (**c**). 200 Hz and (**d**). 20 Hz correspond with the same conditions, whose data is displayed in the The X–Y plots of the cumulative sum of the mean (±standard error) numbers of regenerated axons that advanced into the lesion site as a percent of all the labeled axons proximal to the lesion, The plots show the significant increase in both the extent and distance of axon outgrowth after the CL and the significant increase in the extent but not the distance of axon outgrowth after 20 Hz ES groups. There was no significant increase in either extent or distance of axon outgrowth in the 200 Hz ES group, despite the increased cAMP recorded after 200 Hz. Significant values (*p* < 0.05) are denoted by *s. The scalebar = 250 µm in (**B**). Adapted from [368].

**Figure 19 ijms-25-00665-f019:**
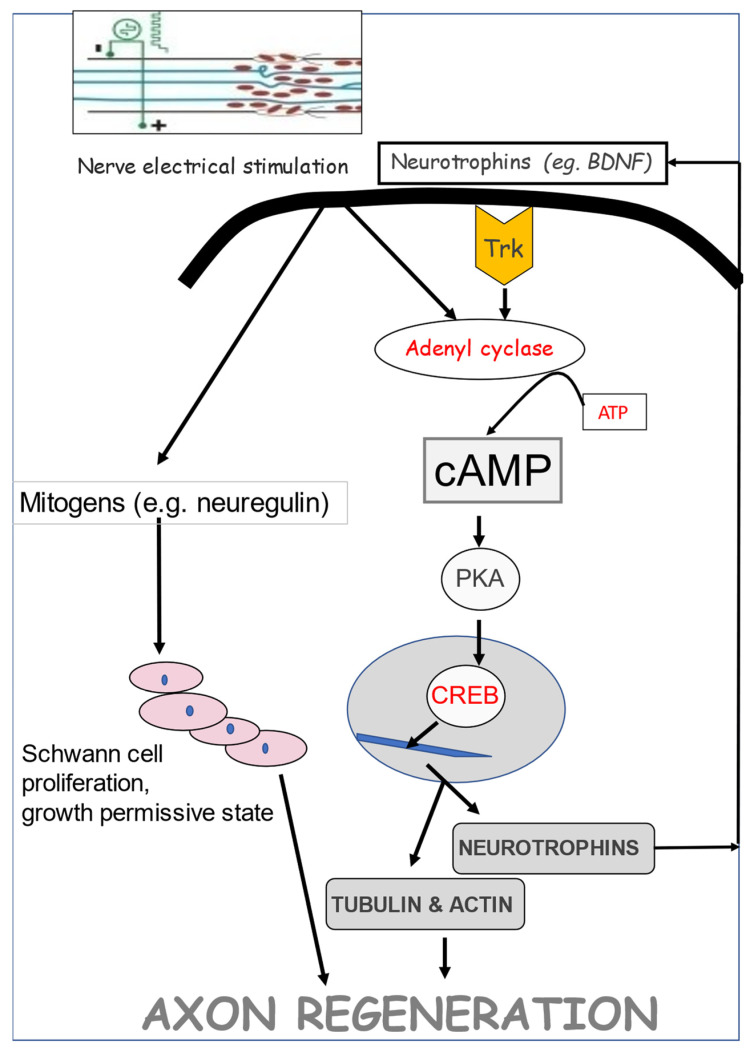
Schematic representation of the intracellular signaling pathways activated by brief (1 h 20 Hz electrical stimulation (ES)). ES increases calcium influx (not shown) and, in turn, activates adenyl cyclase to elevate cAMP levels. Cyclic AMP, in turn, acts via PKA to phosphorylate CREB in the nucleus leading to transcription of proteins, including neurotrophins and the cytoskeletal proteins tubulin and actin that are essential for axonal elongation during regeneration. The neurotrophins also feedback to further increase cAMP (via ERK with transient inhibition of phosphodiesterase 4 to prevent hydrolysis of cAMP to 5′AMP—not shown). Thereby the transcription of regeneration-associated genes is amplified to promote axonal outgrowth. The ES of the injured nerve proximal to the site of nerve repair likely leads to the release of mitogens such as neuregulin that promotes Schwann cell (SC) proliferation and transition from the myelinating to a growth-permissive state. The ES is also effective after chronic nerve injury, presumably by the same mechanism. The growth permissive state of the SCs includes the transcription of neurotrophic factors that likely potentiate axon regeneration via their amplification of neurotrophic factor expression by the neurons. These feedback loops likely aid in promoting accelerated regeneration and target reinnervation.

**Figure 20 ijms-25-00665-f020:**
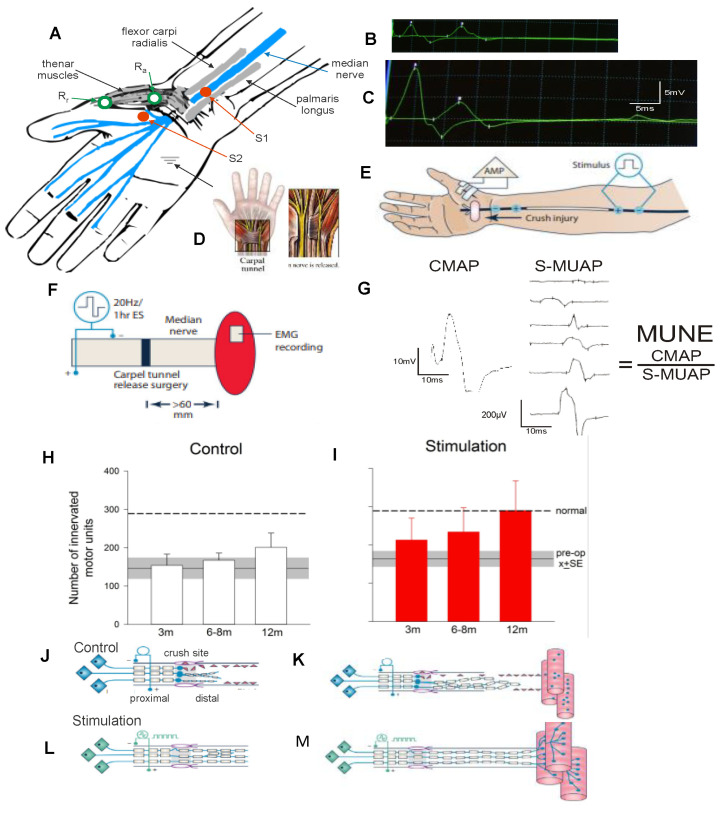
Brief (1 h) 20 Hz electrical stimulation (ES) of injured human nerves accelerates muscle reinnervation. (**A**). Diagrammatic representation of the recording of CMAPs (compound muscle action potentials) in the muscles of the median eminence above and below the ligament of the carpal tunnel. Only patients with severe CTS (carpal tunnel syndrome) with (**B**). reduced numbers of innervated MUs (motor units) exemplified by reduced CMAP amplitudes recorded both above and below the nerve were included in the study of the regenerative effect of ES. (**C**). Patients with a reduced CMAP amplitude recorded below in the wrist but not above the released nerve, evident of conduction block and not MU loss, were excluded. (**D**). The ligament over the median nerve was released (CTRS: carpal tunnel release surgery). (**F**). Immediately after surgery, the median nerve was subjected to ES. (**E**). A maximum (2× threshold) electrical stimulus above the carpal tunnel evoked (**G**). a CMAP and the minimal stimuli evoked all-or-none single MU action potentials (S-MUAP) at points along the median nerve to estimate MU number (MUNE) from the ratio of the CMAP and mean of at least 20 recorded S-MUAPs. Histograms of the mean (+standard error) MUNE recorded before and after CTRS in (**H**). control subjects (without ES, clear histograms) and (**I**). the subjects with stimulation (with ES, closed red histograms). Diagrammatic illustration of **(J**). the sluggish growth of axons across the nerve crush site in the Control subjects that likely accounts for the (**K**). the regenerating nerve fibres not reaching the muscle to increase the numbers of reinnervated MUs, (**L**). axonal growth accelerated across the carpal tunnel crush site by electrical stimulation at 3 months, and (**M**). regenerating nerves progressing through the distal stumps to reinnervated denervated muscle fibres in the thenar eminence by 6 months, accounting for the restoration of the full complement of MUs by 12 months after CTRS and stimulation. Adapted from [416].

## Data Availability

Not applicable.

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
