# Peer review of "Brief Electrical Stimulation Promotes Recovery after Surgical Repair of Injured Peripheral Nerves"

_ijms, 2024, doi:10.3390/ijms25010665_

Round 1

Reviewer 1 Report

Comments and Suggestions for Authors

1.      Author provides a very extensive review however an illustrative overview (flow-chart etc.) about the organization of various details in the review would be more helpful to navigate.

2.      The illustration in the figure makes a point via the logic of 1mm/day growth of the axons, are there clinical basis for that, if so, please cite the relevant literature.

3.      Fig.1 illustration is mentioned twice in the manuscript, I guess it is to emphasize or by error, in any case it is only required once.

4.      The literature where the data figures are appeared first or adopted from, must be cited in the figure legends.

5.      The picture quality (e.g., Fig3a, b illustration) needs more work they are not readable.

6.      Lines 793-95 needs to clarified and rephrased.

7.      One of the avenues which is not extensibly covered is electrical stimulation biases along with the methods to assess the regenerative potential in a clinical setting, this would probably make the review more comprehensive (however left to author’s will).

8.      Author should probably consider mentioning; at what amplitude 20Hz frequency was applied when considered for stimulation.

9.      Fig.11. legends, doesn’t look good.

10.   Author mentioned about the TTX-application, which is not a complete blocker of electrical activity, therefore based on such supporting literature does author propose that only high-frequency-stimulation (i.e.,20Hz and 200Hz) is of importance for regenerative potential pertaining to its acceleration.

11.   Does stimulation affect the BDNF secretion? Is there a linear secretion corresponding to stimulation?

12.   Fig.16. Do DRG-neurons of all ages respond to stimulation equally?

13.   Is there any substantial investigation/or effect found while stimulating the sciatic area locally or via distal stimulation at the disc or at the level of CNS?

14.   Lines 1223-25; Forskolin induced elevation of cAMP is not the only factor that contributes to the regeneration, Forskolin is also shown to potentiate the neurons via its action on vesicle priming proteins, moreover it is a carcinogenic substance, so its therapeutic value is undermined, what is author`s perspective? Please adapt accordingly if necessary.

15.   Lines 1326-30; atrophic state can further be attributed to metabolic state and factors like availability of nutrients at the site of stimulation, isn’t it?

16.   All figures need to be worked out, the content is not clearly visible and lacks original citations that should be preferable mentioned in the figure legend.

17.   Please at your convenience (if possible) simplify the language of your arguments further for wider audience since such an extensive review can be a good introductory resource for many new people in the field.

Comments on the Quality of English Language

Would be nice if it is bit simplified  and the manuscript is more streamlined for its simple flow of ideas.

Author Response

Thank you very much for your review with its helpful comments and suggestions. I have made many revisions that include addressing all your comments and suggestions

  1. Two tables have been created in which

1) I have listed the main titles and the subtitles that should make it much easier for a reader to follow the review as well as to go to the section of interest. It would be very nice if the journal could provide the page numbers to add to this table;

2) I list all the abbreviations that I have used throughout the review as the long names of transcription factors, for example, had made some of the text cumbersome.

  1. I have included in the Figure legend of Figure 1, an explanation of the derivation of the regeneration rate.
  2. I have removed the inadvertent inclusion of a second copy of Figure 1 in the manuscript.
  3. In the Figure legends, I now include a reference to the literature from where the data figures came from.
  4. I have removed the ‘blue’ backgrounds of this now Figure 2 and the figures that follow, to improve their clarity. I have also revised the legend of Figure 2 to provide the necessary information of how the data was collected. This information also impacts on your 7th point.
  5. The paragraph in which the previous lines 793-95 has been shortened as were many paragraphs throughout the manuscript. I believe that, in doing so, I have clarified the information.
  6. Having clarified our methods for calculating the number of reinnervated motor units (MUs) by the ratio of the recorded muscle and MU twitch forces in figure 2, the method for assessing regenerative potential in patients using EMG recordings from reinnervated muscles in Figure 20, should be clearer.  I do point out in lines 498-502, the problems of the use of current clinical measures in providing an accurate assessment of functional recovery.  EMG recordings in these patients might allow a better judgement of regenerative success. There would be fewer potentials recorded by electromyography. This would indicate fewer than normal motor units in the muscle despite the good return of muscle force (due to the enlarged motor units) even when the MRC 5-point scale, for example, indicates that all regenerating nerves reinnervated denervated muscle(s).  
  7. In the legend of Figure 13, have added all the information concerning the details of the methods that were used to stimulate the proximal stump of a microsurgically repaired femoral nerve.
  8. The legends for Figure 11 have been clarified.
  9. Concerning the dose of the TTX applied in our experiments, we selected the dose for the application by establishing it as the dose that completely blocked the conduction of all action potentials. We did not determine whether frequencies of electrical stimulation less than 20 Hz were effective. We had set 20 Hz as the chosen frequency based on the evidence for this frequency being the average firing frequency in motoneurons (Burke, 1991, reference number 409 in the reference list). What was very interesting to us is that this low frequency was effective for sensory as well as motor nerve regeneration.

  1. Unfortunately I do not know the answer to this question.

  1. We have not determined whether young or old neurons respond equally to the electrical stimulation of injured nerves. This is an interesting question. Currently the efficacy of electrical stimulation in young children is being explored. Dr. Greg Borschel has some anecdotal evidence of accelerated recovery in young children. The evidence that I present in Figure 20 is of the data obtained from patients 20 to 86 years old, with a mean (±SD) age of 56 ± 17 years ( Gordon et al., 2010, reference number 416).

  1. An interesting question but my answer is no, to the best of my knowledge.

  1. Again I shortened this section in responding to the request from the reviewing editor that I shorten the review substantially. Indeed, I don’t believe that forskolin has limited therapeutic value, especially for nerve regeneration, despite the positive findings in the laboratory.

  1. I have added ‘and possibly nutrients at the site,’ to line 850, to address you interesting question.

  1. I have removed the ‘blue’ backgrounds of the figures and worked to clarify the contents. I have also revised the legends to the figures to ensure that the methods used and the results obtained are clear. 

  1. Indeed, in my extensive revision of the text with considerable shortening of the text, I have kept a ‘lighter’ tone. Explanation of the methods and results are provided in the legends to the figures.

Finally, I want to thank this reviewer for the many insights into the subject of the review. His comments really aided in my revision of the manuscript.

Reviewer 2 Report

Comments and Suggestions for Authors

In this manuscript, the author provides a thorough and in-depth review of the issue of regeneration of injured peripheral nerves, with a particular focus on recent studies examining the effects of applying brief low-frequency electrical stimulation (ES) or conditional ES (CES) on axonal outgrowth and regeneration. This review encompasses fundamental information about peripheral nerve injury, the factors limiting functional recovery, the effects and limitations of applying exogenous neurotrophic factors, as well as research on how ES accelerates nerve regeneration. For those interested in this field, this review offers an educational and comprehensive resource.

The review also showcases decades of fruitful work in the field of peripheral nerve recovery, which has the potential to significantly enhance the quality of life for patients with peripheral nerve injuries. All my concerns are minor with the majority related to figure presentations and text editing.

Comments

Related to context:

1. In the introduction section, in the second paragraph, I suggest swapping the order of the introduction of ES for injured nerves (line 48-53) and the review of FK506 (line 53-56). So this paragraph talks about drugs and growth factors first and then introduces ES.

2. In section 3.1 “Time- and distance-related decline in nerve regenerative capacity”, I think the only time-related decline in nerve regenerative capacity was discussed. Would it be better to remove the distance-related in the title?

Related to figures and figure captions:

1. In Figure 3: Subpanel labels “A” and “B” were slighted covered by subpanels; In subpanel E, the scale of the white calibration bar was missing; In subpanel F, should remove “E” in the y-axis label. Also in subpanel F, if those red and blue bars were detected numbers of motoneurons, I suggest plotting them as dots on top of the mean bar rather than plotting each realization as an individual bar.

2. In Figure 4: Subpanel C and D, does the red neuron represent regenerative success? In that case, I suggest do not use red text color in “Prolonged axotomy (no target contact)” and “Prolonged SC denervation”. At first read, I found it confusing and thought the color in the title matched the color in the plot. Also, I suggest changing the title of subpanel D to “Prolonged SC denervation further reduces regenerative success”; In the last sentence of the figure caption “The methods used to determine regenerative success are shown in Fig. 1”, I believe it was an incorrect cross reference.

3. In Figure 5: Subpanel B, the scale of the calibration bar was missing; Subpanel C,  resolution too low, axes labels were not readable

4. In Figure 6: Subpanel B,E&F, resolution too low, axes labels were not readable

5. In Figure 8: Subpanel A-F, background colors were not consistent; Subpanel label “E” covered  y-axis label. In the first sentence of its caption, “Exogenous delivery of brain- and glial cell-derived neurotrophic factors, GDNF and BDNF respectively”, should be “..., BDNF and GDNF respectively”

6. In Figure 9: Subpanel C, the scale of the calibration bar was missing;

7. In Figure 10: Subpanel B, y-axis tick labels were missing; Subpanel C, x-axis tick labels were missing, looks like the figure was cropped

8. In Figure 12: Subpanel L, I believe the illustration of ES ‘straightening’ out effect was missing; In Subpanel D,F,N,O, what does b mean? (mu for Muscle and Cu for Cutaneous); Low resolution in subpanel O;

9. In Figure 13, subpanel C, is there a direct control experiment on the impact of α-BDNF on regenerated motoneurons (to support the claim “The numbers of motoneurons that regenerated their axons inappropriately into the sensory branch of the femoral nerve or those that regenerated their axons into both branches were not changed by the infusion of α-BDNF”)?

10. Figure 15 Subpanels B and C were hard to read. Why there were 4 experiments but 5 rows? What were those bars and why they had different colors?

Related to text:

1. There are places with incorrect spacing. I list their line numbers: 29; 161; 193; 194; 200; 259; 277; 329; 343; 352; 378; 596; 603; 604; 615; 665; 681; 697; 712; 719; 740; 859; 860; 875; 891; 947; 959; 974; 1015; 1096; 1125; 1126; 1142; 1144; 1167; 1257; 1291; 1438

2. Punctuation errors (by line numbers): 450; 496; 733; 817; 924

3. Abbreviation issues: In line 99 “DRG” is not defined at its first use; In line 554, 556, 559 and 568, should use a consistent abbreviation (“MyoD” or “MyD”); In line 989 and 990, “ILMO3” or “ILMOS”; Line 1042 define GAG at its first use

4. Cross-reference issue. In section 2 “Peripheral Nerve Injury” subsection titles were 2.1, 2.2, ... In the text, they were referred as 2a, 2b, 2c... (line number:86;369;670;744;811;916;998;1042)

5. I find the title of Section 4 difficult to understand “Neurotrophic Factors to Sustain Their Levels”, sustain whose levels?

6. Line 492 “... by reinnervation by chronically axotomized motoneurons...” Consider changing this to “...through the reinnervation of chronically axotomized motoneurons...”

7. Line 870 “... in straight lines or or take...” duplicated “or”

9. Line 925 “BNDF” should be “BDNF”

Comments on the Quality of English Language

See comment above

Author Response

Thank you for your kind comments on the manuscript. I revised the language in the manuscript substantially in order to shorten the length of the manuscript without sacrificing its content. The shortening was requested by the reviewing editor.

I have addressed all of your comments and requests.  Below are my answers to each of the points that you made.

  1. I have taken your advice to swap the order of the introduction. I have shortened this Introduction section to introduce the administration of drugs succinctly and to immediately state the objectives of the review.

  1. In Section 3.1, the title now excludes mention of the distance related decline. Thank you for this suggestion.

Figures: Thank you for your many suggestions. There were many errors when I transferred the figures to the text. I have done extensive revisions throughout. I will address each in turn.

  1. Figure 3. I have removed the blue background in this and all the figures for more clarity. With regards to F. I have decided to leave it as is because it fits into the figure and gives the reader a perspective of the similarities in the counted number of motoneurons that regenerated their axons into the distal stump in every experiment. This point would not be made by simply displaying the individual experiments as dots.

  1. In what was Figure 4 and is now Figure 3 on Page 16, the Y axis shows the mean numbers + SEs of a. reinnervated motor units (the number of nerves that regenerated and reinnervated the tibialis muscle) and b. motoneurons that regenerated their axons from proximal into distal nerve stumps. I have reworded the legend accordingly.  You are also correct that I cross referenced Figure 1 when I meant to cross-reference the then Figure 3 (which is now Figure 2)

  1. Figure 5. I apologise for this missing information. I have now included the value of the scale bar at the bottom right of Figure 5B.  I have also revised the Figure 5C to clarify the time-dependent decay in  the in situ hybridization

  1. Figure 6. I believe that the resolution has been improved.

  1. Figure 8. The resolution has been improved by eliminating the blue background and the placement of the label ‘E’ has been corrected. The first sentence in the legend has been corrected.

  1. Figure Again, I must apologize for my tardiness with the scale bars in the figures. The scale bar has been included in the Figure legend.

  1. I have explained in the legend of Figure 10 that the ‘values of the tick marks on the Y-axis in A and B are the same’.

  1. Figure 8. I have revised the sentence describing the subpanel L to “Diagrammatic illustrations of the axons proximal to the site of a surgical repair if the transected nerve and the contorted pattern of progress of regenerating axons across a suture site”

  1. Figure 13 (was Figure 12). Although we have some of the control data that showed no effect of the BDNF antibody on the regeneration of the unstimulated femoral nerves, there are not sufficient data to include in the figure. We refer to it in the second last sentence of the legend.

  1. Figure 15. I have now revised this figure and its legend for legibility and clarity.  I have also deleted the micrographs of the in situ hybridization data that were not referred to.  

Round 2

Reviewer 1 Report

Comments and Suggestions for Authors

1.      Author have addressed majority of the concerns however there is still work required in terms of enhancing the figure quality.

2.      Fig.5a,7a are repetition from Fig.4 and are not required, however subjected to author’s taste.

Comments on the Quality of English Language

       There are still some structural changes e.g., sentence structure and paraphrasing required e.g lines 228-29 etc….Please do the needful!

Author Response

   Fig.5a,7a are repetition from Fig.4 and are not required, however subjected to author’s taste. Response I have decided to keep 7a.

  There are still some structural changes e.g., sentence structure and paraphrasing required e.g lines 228-29 etc….Please do the needful! Response Thank you for pointing out the structural changes necessary. I have read through the manuscript carefully and made several of these changes. 

Reviewer 2 Report

Comments and Suggestions for Authors

All the comments have been addressed. No further requirements.

Author Response

Thank you